# A Few Moments Please: Scalable Graphon Learning via Moment Matching

**Reza Ramezanpour**
Rice University
rr68@rice.edu

**Victor M. Tenorio**
King Juan Carlos University
victor.tenorio@urjc.es

**Antonio G. Marques**
King Juan Carlos University
antonio.garcia.marques@urjc.es

**Ashutosh Sabharwal**
Rice University
ashu@rice.edu

**Santiago Segarra**
Rice University
segarra@rice.edu

## Abstract

Graphons, as limit objects of dense graph sequences, play a central role in the statistical analysis of network data. However, existing graphon estimation methods often struggle with scalability to large networks and resolution-independent approximation, due to their reliance on estimating latent variables or costly metrics such as the Gromov-Wasserstein distance. In this work, we propose a novel, scalable graphon estimator that directly recovers the graphon via moment matching, leveraging implicit neural representations (INRs). Our approach avoids latent variable modeling by training an INR–mapping coordinates to graphon values–to match empirical subgraph counts (i.e., moments) from observed graphs. This direct estimation mechanism yields a polynomial-time solution and crucially sidesteps the combinatorial complexity of Gromov-Wasserstein optimization. Building on foundational results, we establish a theoretical guarantee: when the observed subgraph motifs sufficiently represent those of the true graphon (a condition met with sufficiently large or numerous graph samples), the estimated graphon achieves a provable upper bound in cut distance from the ground truth. Additionally, we introduce MomentMixup, a data augmentation technique that performs mixup in the moment space to enhance graphon-based learning. Our graphon estimation method achieves strong empirical performance–demonstrating high accuracy on small graphs and superior computational efficiency on large graphs–outperforming state-of-the-art scalable estimators in 75% of benchmark settings and matching them in the remaining cases. Furthermore, MomentMixup demonstrated improved graph classification accuracy on the majority of our benchmarks.

## 1 Introduction

Networks are fundamental structures for representing complex relational data across diverse domains, from social interactions and biological systems to technological infrastructures [10, 31]. Understanding the underlying principles governing these networks is crucial for tasks such as link prediction [26], community detection [30], and, more broadly, node or graph classification [25]. Graphons, or graph limits, have emerged as a powerful mathematical framework for capturing the asymptotic structure of sequences of dense graphs [19, 18, 5, 2]. They provide a continuous, generative model for graphs, enabling principled statistical analysis and offering a canonical representation for large networks. Graphons have been successfully applied to derive controllers for large networks [12], to understand network games with many actors [23], to perform data augmentation in graph settings [21, 13], and

---

*Source Code: https://github.com/rezar76/Graphon-Moment-Matching

39th Conference on Neural Information Processing Systems (NeurIPS 2025).

to aid in the topology inference of partially observed graphs [28, 20]. As such, developing accurate and efficient methods for estimating graphons from observed network data is a central problem in network science and machine learning.

Estimating graphons from finite, potentially noisy graph observations presents significant challenges. Many existing approaches suffer from computational scalability issues when applied to large networks [7, 35]. Furthermore, their resolution is limited by the size of the sample graphs, and obtaining a resolution-free approximation of the underlying continuous graphon can be difficult [7]. For instance, implicit neural representations (INRs) have been explored for graphon estimation due to their ability to model continuous functions [34]. However, estimating the latent variables of the nodes to train the INRs remains a challenge, and oftentimes the literature resorts to computationally demanding optimal transport-inspired losses, like the Gromov-Wasserstein (GW) distance for optimization. While recent scalable methods have made progress [3], there remains a need for estimators that combine high accuracy, computational efficiency, and direct graphon recovery without complex intermediate steps.

In this paper, we introduce a novel and scalable approach for graphon estimation via moment matching, designed to overcome these prevalent limitations. Our method directly recovers the graphon by leveraging subgraph counts (graph moments) from observed data, thereby bypassing the need for latent variables and their associated complexities. We represent the graphon using an INR, a continuous function parameterized by a neural network that maps coordinates in $[0,1]^2$ to the corresponding graphon value. The parameters of this INR are learned by minimizing the discrepancy between the moments derived from the INR and the empirical moments computed from the input graph(s). This direct recovery strategy, crucially, leads to a polynomial-time estimation algorithm and does not rely on combinatorial GW distances, distinguishing it from approaches like IGNR [34]. Our approach is underpinned by a theoretical result, building upon foundational work on convergent graph sequences [5], which establishes that if the motifs (subgraph patterns) in the observed graph data sufficiently represent the motifs present in the true underlying graphon–a condition met with sufficiently large or numerous graph samples–then the cut distance between the estimated and true graphons is provably upper bounded. Additionally, we propose MomentMixup, a novel data augmentation technique that operates by interpolating graph moments between classes and then learning the corresponding mixed graphons, offering an improvement over existing mixup strategies in the graphon domain [21, 13].

Our contributions are threefold:

1. We propose MomentNet, a scalable graphon estimator based on moment matching with INR, offering a resolution-free and estimation recovery mechanism.

2. We provide a theoretical guarantee linking the fidelity of motif representation in observed data to the estimation accuracy in terms of cut distance.

3. We introduce MomentMixup, an effective data augmentation method in the moment space for graphon-based learning tasks.

The remainder of this paper is structured as follows: Section 2 presents the necessary background concepts and related works. Section 3 details our moment-matching INR approach for graphon estimation, including its theoretical characterization. Section 4 introduces MomentMixup, our approach for data augmentation in graph classification tasks. Section 5 presents our comprehensive empirical evaluations in both synthetic graphon estimation and data augmentation for graph classification. Finally, Section 6 concludes the paper and discusses future directions.

## 2 Background, Related Works and Problem Formulation

In this section, we introduce the foundational concepts of graphons, motif densities, INRs for graphon estimation, and mixup for data augmentation. We also formally state the graphon estimation problem addressed in this paper. In all these topics, we provide a summary of the literature, although a detailed discussion of related works can be found in Appendix A.

**Graphons**   A graphon, short for "graph function," is a fundamental concept in the theory of graph limits, serving as a limit object for sequences of dense graphs [18, 5]. Formally, a graphon $W$ is a symmetric measurable function $W : [0,1]^2 \rightarrow [0,1]$. Intuitively, the unit interval $[0,1]$ can be thought of as a latent space for the graph nodes. For any two points $x, y \in [0,1]$ (representing latent

positions), the value $W(x, y)$ represents the probability of an edge forming between nodes associated with these latent positions.

A random graph $G_n(W)$ with $n$ nodes can be generated from a graphon $W$ by sampling $n$ i.i.d. latent positions $\eta_1, \eta_2, \ldots, \eta_n \sim \mathcal{U}[0, 1]$ and, for each pair of distinct nodes $(i, j)$ with $1 \leq i < j \leq n$, an edge $(i, j)$ is included in $G_n(W)$ independently with probability $W(\eta_i, \eta_j)$. Graphons are inherently invariant to permutations of node labels in the generated graphs, meaning that different orderings of the latent positions $\eta_i$ that preserve their relative positions in $[0, 1]$ (or more formally, measure-preserving bijections of $[0, 1]$) lead to equivalent graphon representations. The natural distance metric capturing this invariance is the cut distance [5].

**Motif Densities from Graphons**  A key property of graphons is their ability to characterize the expected density of small subgraphs, often called motifs [5, 18]. For a simple graph $F$ (the motif), whose node and edge set are represented by $\mathcal{V}_F$ and $\mathcal{E}_F$, respectively, with $k = |\mathcal{V}_F|$, its homomorphism density in a graphon $W$, denoted $t(F, W)$, is defined as

$$t(F, W) = \int_{[0,1]^k} \prod_{(i,j) \in \mathcal{E}_F} W(\eta_i, \eta_j) \prod_{l \in \mathcal{V}_F} d\eta_l. \tag{1}$$

This integral represents the probability that $k$ randomly chosen latent positions from $[0, 1]$ induce a subgraph homomorphic to $F$ according to the edge probabilities defined by $W$. For a sufficiently large graph $G$ sampled from $W$, the empirical count of motif $F$ in $G$, normalized appropriately, converges to $t(F, W)$. Thus, empirical motif densities from observed graphs can serve as estimators for the true motif densities of the underlying graphon. The set of all such motif densities $\{t(F, W)\}_{F \in \mathcal{F}}$ (for some collection of motifs $\mathcal{F}$) is often referred to as the moment vector of the graphon [4]. We also introduce the induced motif densities as follows

$$t'(F, W) = \int_{[0,1]^k} \prod_{(i,j) \in \mathcal{E}_F} W(\eta_i, \eta_j) \prod_{(i,j) \notin \mathcal{E}_F} (1 - W(\eta_i, \eta_j)) \prod_{l \in \mathcal{V}_F} d\eta_i. \tag{2}$$

This formulation for induced motif density, $t'(F, W)$, specifically counts instances where the motif $F$ appears in $W$ with an *exact* structural match. This means it accounts for both the required presence of edges specified in $F$ and the required *absence* of edges between the motif's vertices that are not in $F$. In contrast, a non-induced (or homomorphism) density $t(F, W)$ only requires the presence of edges from $F$ in $W$, without any assumption of the value of the graphon associated with pairs of nodes not linked by an edge.

**Implicit Neural Representations for Graphon Estimation**  An INR can effectively model a graphon by learning it as a continuous function [34, 3]. In this setup, the INR, typically a neural network $f_\theta : [0, 1]^2 \rightarrow [0, 1]$, is trained to take pairs of latent node coordinates $(\eta_i, \eta_j)$ from a continuous space as input, where $\eta_i$ and $\eta_j$ represent the latent positions associated with entities $i$ and $j$. Its output is the predicted value of the graphon $f_\theta(\eta_i, \eta_j) = \hat{W}(\eta_i, \eta_j)$, representing the probability of an edge existing between these two latent positions. The network $f_\theta$ learns this mapping from observed samples, which could be $((\eta_{i_l}, \eta_{j_l}), W(\eta_{i_l}, \eta_{j_l}))$ pairs derived from a large graph or a target graphon function, for a set of sample indices $l$. Crucially, because $f_\theta$ learns a continuous function over the entire input coordinate space defined by $\eta_.$, the resulting graphon representation is inherently resolution-free. This means it can determine the edge probability for any arbitrary pair of latent coordinates $(\eta_i, \eta_j)$, allowing for the generation or analysis of graph structures at any desired level of detail or scale without being tied to a fixed number of nodes or a specific discretization.

**Mixup for Data Augmentation**  The core idea of Mixup [40] is to generate synthetic training examples by taking convex combinations of pairs of existing samples and their corresponding labels. Given two input samples $x_i$ and $x_j$ with their respective labels $y_i$ and $y_j$, a new synthetic sample $(\tilde{x}, \tilde{y})$ is created as $\tilde{x} = \lambda x_i + (1 - \lambda)x_j$, $\tilde{y} = \lambda y_i + (1 - \lambda)y_j$. where $\lambda \in [0, 1]$ is a mixing coefficient. This encourages the model to behave linearly in-between training examples, leading to smoother decision boundaries and improved generalization.

Applying Mixup directly to graph-structured data presents challenges because graphs are not inherently Euclidean objects. To perform Mixup for graphs, one typically first maps the graphs into a suitable Euclidean representation [21, 13]. For example, GraphMAD [21] maps the graphs to a latent

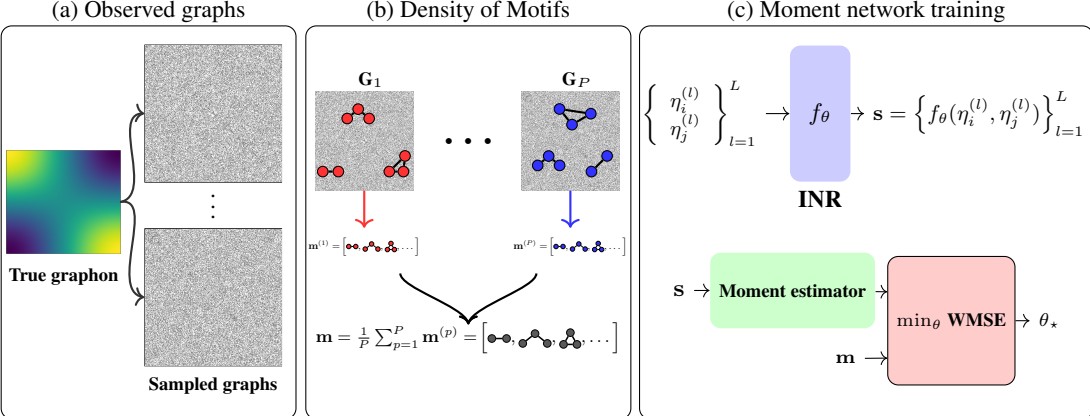

Figure 1: Graphon estimation pipeline: observed graphs lead to motif frequency extraction and INR-based recovery.

space and performs nonlinear mixup, while G-Mixup [13] performs mixup in the graphon domain. Once graphs $G_i$ and $G_j$ are available as Euclidean representations $\mathbf{z}_i$ and $\mathbf{z}_j$ respectively, a mixed representation $\tilde{\mathbf{z}} = \lambda \mathbf{z}_i + (1 - \lambda)\mathbf{z}_j$ can be computed. The subsequent step, which can be non-trivial, is to generate a new graph $\tilde{G}$ from this mixed representation $\tilde{\mathbf{z}}$ that can be used for training a graph classification model.

**Problem Formulation.** The primary problem addressed in the graphon estimation literature, and in this work, is to recover the underlying graphon $W^*$ given one or more observed graphs.

**Problem 1** (Graphon Estimation). *Given a set of observed graphs $\mathcal{G} = \{G_1, G_2, \ldots, G_P\}$, where each $G_p$ has $n_p$ vertices and is assumed to be sampled (conditionally independently) from an unknown true graphon $W^*$, i.e., $G_p \sim G_{n_p}(W^*)$, the goal is to estimate $W^*$.*

In the literature, early methods aimed at solving Problem 1 by means of histogram estimators and stochastic block models [5, 18, 6, 1, 11]. Other non-parametric approaches, like Universal Singular Value Thresholding (USVT) [7], aimed to recover underlying network structures but often faced computational or resolution limitations. More recent scalable techniques include those using INRs. For instance, IGNR [34] often leverages GW distances [24, 37, 35] for alignment, while methods like SIGL [3] further advance INR-based estimation.

Our work proposes a novel method for solving Problem 1 by directly learning an INR to match empirical moments (subgraph counts) from the observed graph(s), thereby bypassing latent variables and computationally expensive metric optimizations. Moreover, we leverage our proposed solution to Problem 1 to design MomentMixup, a novel mixup strategy for graph data augmentation. MomentMixup performs mixup in the space of empirical moments, offering a novel way to generate augmented graph data informed by the underlying generative structure.

## 3 Moment Matching Neural Network (MomentNet)

In the following subsections, we introduce our proposed method, **MomentNet**, for learning the graphon. We also provide the fundamental theorem upon which our model is built.

### 3.1 Methodology

We explain the two steps in our method to estimate the graphon $W$ given the set of sampled graphs denoted by $\mathcal{G} = \{G_p\}_{p=1}^{P}$. A schematic view of our method is presented in Figure 1.

**Step 1: Computing density of motifs.** For each graph in our dataset $\mathcal{G}$, we count the occurrences of specific motifs. The density of an identified motif is then calculated as the ratio of its observed count to the total number of possible ways that particular motif could appear in a graph of the same size.

We use the ORCA method [15] to count the number of graphlets in the graph, and then we convert the graphlet count into motif counts. This aggregation is needed because our analysis cares only about how often each subgraph pattern appears in total, not about the exact placement of individual nodes within those patterns; see Fig. 1 in [15] for an illustration. We parallelize the use of the ORCA method for computing motif counts across the graphs in our dataset, thereby gaining a significant speed-up in processing time. ORCA can count motifs with up to five nodes, and its method can be extended to handle larger motifs. Once these motif-based statistics are calculated from the graphs, we no longer use the graphs themselves for subsequent steps. This approach significantly reduces computational overhead. Mathematically, we consider a set $\mathcal{F}$ of $|\mathcal{F}|$ distinct motif types. For each graph $G_p$ in our dataset, its empirical motif density vector is $\mathbf{m}^{(p)} \in \mathbb{R}^{|\mathcal{F}|}$. The overall motif density vector $\mathbf{m}$ for the dataset is currently computed as the simple average:

$$\mathbf{m} = \frac{1}{P} \sum_{p=1}^{P} \mathbf{m}^{(p)}. \tag{3}$$

While Eq. (3) treats each graph equally, a more general approach could involve a weighted average, $\mathbf{m}_w = \sum_{p=1}^{P} w_p \mathbf{m}^{(p)}$ (where $w_p \geq 0$ and $\sum_{p=1}^{P} w_p = 1$). Such weights $w_p$ could, for example, depend on graph properties like size ($n_p$), potentially giving more influence to larger graphs, which might yield more stable density estimates. Our present work employs the simple average, with the exploration of weighted schemes as a potential future refinement.

**Step 2: Training the Moment network** The moment network is defined as a combination of INR with a moment-based loss function. This step consists of three components described as follows:

1. **INR**: Our methodology employs an INR $f_\theta$ to model the graphon, that receives the latent coordinates $(\eta_i, \eta_j)$ and outputs the estimated graphon value $\hat{W}_\theta(\eta_i, \eta_j)$, as explained in Section 2.

2. **Moment estimator**: With the graphon estimated by the INR function $f_\theta$, we can compute the induced motif density for any given motif $F$. This is achieved by substituting $\hat{W}_\theta$ in place of $W$ in (2). Since we can not compute the integral directly, we approximate it using Monte Carlo integration techniques. By generating a sufficient number of random samples $L$ from the distribution induced by the graphon, we can estimate the integral. More precisely, we sample $L$ samples of $k$ latent coordinates $\eta_1^{(l)}, \ldots, \eta_k^{(l)}$, where $\eta_i^{(l)} \sim \mathcal{U}[0,1]$. Then we estimate $t'(F, \hat{W}_\theta)$ as

$$\hat{t}'(F, \hat{W}_\theta) = \frac{1}{L} \sum_{l=1}^{L} \left[ \prod_{(i,j)\in\mathcal{E}_F} \hat{W}_\theta(\eta_i^{(l)}, \eta_j^{(l)}) \prod_{(i,j)\notin\mathcal{E}_F} (1 - \hat{W}_\theta(\eta_i^{(l)}, \eta_j^{(l)})) \right] \tag{4}$$

The Monte Carlo estimator $\hat{t}'(F, \hat{W}_\theta)$ is differentiable with respect to the INR parameters $\theta$. Since the INR $f_\theta$ is a neural network parameterized by $\theta$ (and thus differentiable with respect to $\theta$), the estimator $\hat{t}'(F, \hat{W}_\theta)$, which is constructed as an average of terms derived from $f_\theta$ outputs at fixed sample points $\boldsymbol{\eta}^{(l)}$, is consequently also differentiable with respect to $\theta$. This characteristic is vital as it allows the use of gradient-based optimization algorithms to train the INR parameters $\theta$ when this estimator is incorporated into a loss function. A proof of unbiasedness for this approach, i.e., showing that $\mathbb{E}[\hat{t}'(F, \hat{W}_\theta)] = t'(F, \hat{W}_\theta)$, is provided in Appendix D. The vector of estimated moments (e.g., motif densities) derived from the INR outputs is denoted as $\hat{\mathbf{m}}(\theta) = \left[ \hat{t}'(F_1, \hat{W}_\theta), \hat{t}'(F_2, \hat{W}_\theta), \ldots, \hat{t}'(F_{|\mathcal{F}|}, \hat{W}_\theta) \right]^\top$.

3. **WMSE**: We use weighted mean squared error as a loss function to train our INR. Given the empirical moment vector $\mathbf{m}$, based on sampled graphs and computed using Eq. (3), and the estimated moments based on the INR as $\hat{\mathbf{m}}(\theta)$, the loss function is

$$L(\theta) = \sum_{i=1}^{|\mathcal{F}|} w_i \left( m_i - \hat{m}_i(\theta) \right)^2. \tag{5}$$

In our experiments, we adjust the importance of different factors by assigning weights ($w_i$). We calculate these weights as the inverse of how strong each factor ($m_i$) appears in our data

$(w_i = \frac{1}{m_i})$. This weighting method balances the impact of each moment, preventing the most frequent ones (larger $m_i$) from having a large effect on the learning process.

The training process described above, optimizing the parameters $\theta$ of the INR $f_\theta$ to minimize the weighted mean squared error between empirical and estimated motif densities, yields our final graphon estimate $\hat{W}_\theta = f_\theta$. This estimated graphon is inherently scale-free due to the continuous nature of the INR. Furthermore, the entire estimation procedure operates in polynomial time with respect to the number of nodes and motifs considered. A detailed complexity analysis is provided in Appendix E.

## 3.2 Theoretical characterization

We present our main theorem bounding the cut distance between the true graphon $W^*$ and the graphon $\hat{W}_\theta$ estimated by our proposed INR. This result combines insights from the concentration of empirical motif densities in the $G_n(W)$ model [5] with the inverse counting lemma relating motif distances to cut distance, and an assumption about the neural network's ability to approximate empirical motif densities. Supporting lemmas and the proof of this theorem are provided in Appendix B.

Let $G_1, \ldots, G_p$ be $P$ graphs, each with $n$ vertices, sampled independently from the graphon model $G_n(W^*)$ according to the graphon $W^*$. The **empirical motif density** of $F$ based on these samples is $\bar{t}(F, W^*) = \frac{1}{P} \sum_{p=1}^{P} t(F, G_p)$, where in a slight abuse of notation we denote by $t(F, G_p)$ the motifs densities computed from the motif counts of graph $G_p$.

We consider an INR $f_\theta$ with parameters $\theta$, whose estimated graphon is denoted by $\hat{W}_\theta$. The motif densities corresponding to this estimated graphon are denoted by $\hat{t}_\theta(F, \hat{W}_\theta)$. The INR is trained to directly output $\hat{t}_\theta(F, \hat{W}_\theta)$ values that approximate the empirical densities $\bar{t}(F, W^*)$. Also, let $\mathcal{F}_k$ denote the set of all non-isomorphic simple graphs with exactly $k$ vertices and let $N_k = |\mathcal{F}_k|$ be the number of such graphs. As a preliminary step, we formalize the performance requirement we use to characterize our neural network next (see Appendix B.2 for a justification).

**Assumption 1** (Neural Network Approximation Capability). *The parameters $\theta$ of the INR $\hat{W}_\theta$ are obtained such that for a fixed approximation error $\epsilon_a > 0$, the estimated motif densities $\hat{t}_\theta(F, \hat{W}_\theta)$ satisfy*

$$|\hat{t}_\theta(F, \hat{W}_\theta) - \bar{t}(F, W^*)| < \epsilon_a, \quad \text{for all } F \in \mathcal{F}_k. \tag{6}$$

With the previous definitions, and those of Lemma 2 in Appendix B, we are in a position to present our main result, stated in Theorem 1.

**Theorem 1** (Cut Distance Bound for INR Estimated Graphons). *Let $\epsilon_a > 0$ be the approximation error achieved by the network as stated in Assumption 1, and $\delta_M = 3^{-k^2}$ be the motif deviation threshold. Assume $n > \frac{k(k-1)}{\delta_M}$ and*

$$N_k \cdot 2 \exp\left( -\frac{Pn}{4k^2} \left( \frac{\delta_M}{2} - \frac{k(k-1)}{2n} \right)^2 \right) < \zeta, \tag{7}$$

*where $\zeta > 0$ is a desired confidence level. Then, with probability at least $1 - \zeta$, the cut distance between the neural network estimated graphon $\hat{W}_\theta$ and the true graphon $W^*$ is bounded by $\eta = \frac{22C}{\sqrt{\log_2 k}}$, with $C = \max\{1, \|W_1\|_\infty, \|W_2\|_\infty\}$, as*

$$d_{cut}(\hat{W}_\theta, W^*) < \eta. \tag{8}$$

A detailed proof of Theorem 1, along with necessary definitions and supporting lemmas, can be found in Appendix B. This result demonstrates that if the INR can accurately approximate the empirical motif densities (Assumption 1), and if enough data (characterized by $P$ and $n$) is available to ensure the empirical motif densities are close to the true graphon motifs (Lemma 1), then the estimated graphon is likely to be close to the true graphon in cut distance.

Although condition (7) may seem restrictive, note that (i) it decays exponentially with the number of graphs $P$ and their size $n$ considered, so it can be made arbitrarily small by considering larger datasets and (ii) although it increases with $k$ (and therefore with $N_k$), the size of the subgraphs considered $k$ is usually small (up to 5 nodes at most).

## 4 Moment Mixup

Data augmentation is a crucial technique in machine learning, particularly in domains like graph learning, where labeled data can be scarce or expensive to obtain [9]. By synthetically expanding the training dataset with new, plausible examples, data augmentation helps to improve model generalization, reduce overfitting, and enhance robustness. In the context of graph learning, developing effective augmentation strategies is challenging due to the complex, non-Euclidean nature of graph data, where direct analogies to image or text augmentation methods are not always feasible.

In this section, we introduce MomentMixup, a novel approach for data augmentation in graph learning. The process begins by generating novel moment profiles through convex combinations of moment vectors, where each vector $\mathbf{m}_k$ is derived from sampled graphs belonging to a distinct graph class (e.g., $\mathbf{m}_{\text{new}} = \sum \alpha_k \mathbf{m}_k$, with $\alpha_k \geq 0, \sum \alpha_k = 1$). This interpolated moment vector, $\mathbf{m}_{\text{new}}$, is then used as the input to MomentNet, which subsequently defines a new graphon distribution, $W_{\text{new}}(\eta_i, \eta_j)$, consistent with these synthesized moments. Finally, new graphs are sampled from $W_{\text{new}}(\eta_i, \eta_j)$ and integrated into the training set. The pseudocode of MomentMixup is provided in Algorithm 1 in Appendix G.

**Proposition 1.** *A convex combination of graphons is not equivalent to the corresponding convex combination of their vectors of moments, with the exception of the edge density moment.*

Proof of Proposition 1 using a counterexample is provided in Appendix F. MomentMixup is developed based on the key insight from Proposition 1. This foundational understanding distinguishes MomentMixup and offers it as an alternative to existing methods like G-Mixup [13]. The core intuition underpinning MomentMixup is that newly generated graph samples should exhibit clear structural proximity to a specific class (i.e., similar motif counts), thereby ensuring the augmented data reinforces class-specific structural characteristics. We contend that this particular intuition, that a generated sample is structurally close to a target class, may not always be reliably achieved through G-Mixup's graphon interpolation strategy because of Proposition 1. Furthermore, a detailed reproducibility analysis was unable to substantiate the original paper's claims regarding G-Mixup's superiority over other data augmentation methods [22].

## 5 Numerical Experiments

In this section, we evaluate the performance of **MomentNet** and **MomentMixup** using various synthetic and real-world datasets widely used in the literature. The primary deep learning components of our experiments were executed on an Nvidia A100 GPU. Separately, empirical graph moments were computed using the ORCA toolkit [15], with its execution parallelized across an AMD EPYC 7742 64-Core Processor.

### 5.1 MomentNet Evaluation

To comprehensively evaluate our proposed **MomentNet**, we focus on two critical dimensions. First, we examine its effectiveness in the primary task of graphon estimation, determining how accurately it can capture the underlying distribution of graphs. Second, we address the practical applicability of our model by testing its scalability. This involves assessing its performance and runtime when applied to both large graphs (high number of nodes) and collections of smaller graphs, which are crucial considerations for real-world applications. We use $L = 20000$, which is the number of samples to compute the density of moments of MomentNet using Eq. 4 in both experiments.

### 5.1.1 Graphon Estimation

We use the 13 graphon distributions used in the literature [3, 34]. The list of graphon distributions with their plot is provided in Appendix G. To build our experimental dataset, we adopt the graph generation approach utilized in [34]. From each graphon, we then generate 10 distinct graphs of varying sizes, specifically containing $\{75, 100, \ldots, 275, 300\}$ nodes respectively. We treated the INR architecture as a hyperparameter to account for function complexity, noting that a simple one-layer MLP [17] with 64 neurons sufficed for non-complex graphons, while more complex cases (like Stochastic Block Models) required architectures such as SIREN [29] to accurately represent high-frequency details. This reflects the known limitations of modeling complex functions with small networks;

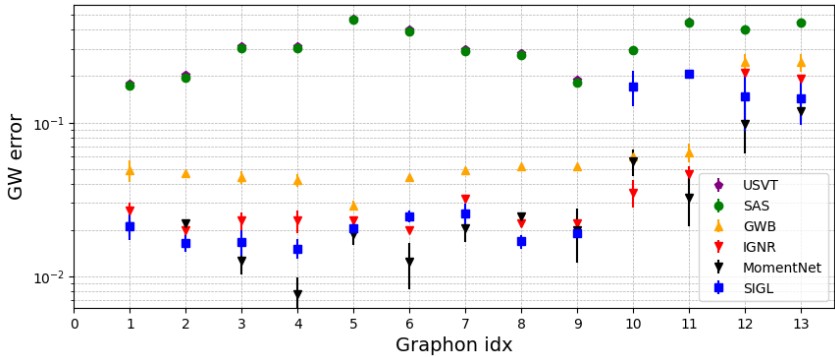

(a) Performance comparison of MomentNet compared to other graphon estimation approaches.

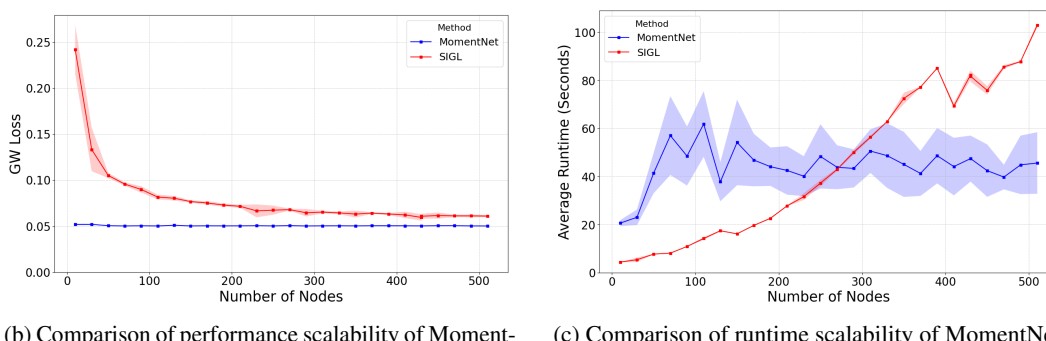

(b) Comparison of performance scalability of Moment-Net with SIGL.

(c) Comparison of runtime scalability of MomentNet with SIGL.

Figure 2: Overall comparison of MomentNet performance and scalability.

the specific best-performing architecture for each graphon is provided in the supplementary code repository. For comparison, following INR training, we generate the graphon using 1000 uniformly sampled equidistant latent variables over the interval $[0, 1]$. The GW distance [36] is then computed between this estimate and the ground truth graphon. For our method, we implemented the same steps described in section 3, considering the motifs provided in Fig. 5. To evaluate the performance of our graphon estimation, we benchmark it against several established baseline methods. These include universal singular value thresholding (USVT) [7], sorting-and-smoothing (SAS) [6], implicit graph neural representation (IGNR) [34], Graph-Wasserstein barycenters (GWB) [35], and scalable implicit graphon learning (SIGL). For a consistent comparison, graphons estimated by the IGNR and SIGL baselines are sampled at a resolution of 1000, mirroring our own evaluation protocol. Furthermore, for SAS and USVT, we zero-pad the adjacency matrices of the observed graphs to this target resolution of 1000 before their respective graphon estimation procedures are applied; this input processing strategy is similar to those employed in [3, 34].

The results are provided in Fig. 2a. Based on the GW loss, our method outperforms the scalable state-of-the-art approach in 9 out of 13 graphons. Notably, our approach achieved superior results for graphons 10 and 11, where the state-of-the-art baseline (SIGL) struggled more. This difficulty might stem from the specific structures of graphons 10 and 11, which can challenge SIGL's reliance on accurately learning latent variables for its GNN-based node ordering and subsequent graphon estimation. Alongside the GW loss comparison, we assessed our graphon estimation via centrality measures, and the findings, detailed in Appendix J, affirmed our method's performance.

### 5.1.2 Scalability Evaluation

For experimental evaluations, we use the graphon $W(\eta_i, \eta_j) = 0.5 + 0.1 \cos(\pi\eta_i) \cos(\pi\eta_j)$ (Figure 1) for generating graph instances across multiple independent realizations for each node size $n \in \{10, \ldots, 510\}$. In each realization, 10 graphs of size $n$ are generated; MomentNet's target motif counts are averaged from these, while SIGL processes them according to its methodology.

Reported performance metrics are averaged over these realizations, allowing methods to leverage a comprehensive set of samples.

The estimation error results (Figure 2b) show that MomentNet achieves strong performance, with error decreasing as $n$ increases. By leveraging multiple graph instances, MomentNet demonstrates near-optimal performance even on relatively small graphs, attributed to more accurate motif density estimation, aligning with theory. In contrast, SIGL's error, while node-dependent, does not substantially improve from multiple graph instances, offering only slight gains for small graphs, resulting in inferior overall performance. A potential explanation is SIGL's reliance on accurate latent variable estimation. The specific graphon $W(\eta_i, \eta_j) = 0.5 + 0.1\cos(\pi\eta_i)\cos(\pi\eta_j)$ (Figure 1) makes this challenging, as its construction ensures edge probabilities near $0.5$, leading to high variance ($W(1 - W)$ near maximum). This high variance can obscure latent structure. MomentNet's averaging directly reduces density estimate variance. However, for SIGL, if each of the 10 graphs individually fails to resolve latent positions due to high variance, more such graphs may not overcome this limitation as effectively as methods that directly average structural features.

Regarding runtime (Figure 2c), MomentNet's average runtime, despite variance, scales more favorably with increasing nodes compared to SIGL, showing a clear advantage for $n > 300$. The variance in MomentNet's runtime is due to its early stopping criteria (see Appendix E for detailed complexity analysis). Further experimental results are provided in Appendix K.

### 5.1.3 Ablation Study: Choice of Moments

The robustness of MomentNet is demonstrated by its strong performance using a relatively small, fixed set of motifs. We conducted an ablation study to investigate the impact of moment selection, with results for Graphons 2 and 4 (from Table 7) presented in Table 1. A key finding is that while performance generally improves as more motifs are added (indicated by lower GW distance), this trend is not strictly monotonic. We observed minor performance dips, for instance, after incorporating the seventh and eighth motifs for Graphon 2, a behavior also seen with Graphon 4.

This non-monotonicity suggests that not all motifs contribute equally; some higher-order motifs may introduce statistical noise, as they are often rare and thus typically require more samples for accurate approximation. Critically, these fluctuations are minor within a strong overall trend, and adding motifs beyond the first six provides no significant advantage. Our experiments affirm that a practical and powerful graphon estimation can be achieved using a fixed set containing all motifs up to a small node count $k$. While an adaptive motif selection strategy remains a valuable avenue for future exploration, the current approach is robust and highly effective.

Table 1: Ablation of motif count vs. GW distance (Avg $\pm$ Std). Motif count indicates the cumulative number of motifs used (up to 15, which is all motifs of size $\leq 5$).

| Motif Count | 1 | 2 | 3 | 4 | 5 | 6 | 7 | 8 | 9 | 10 | 11 | 12 | 13 | 14 | 15 |
|---|---|---|---|---|---|---|---|---|---|---|---|---|---|---|---|
| **GW Graphon 2** | .089 | .100 | .036 | .031 | .029 | .021 | .024 | .027 | .026 | .022 | .023 | .020 | .020 | .018 | .020 |
| **Std Graphon 2** | .012 | .015 | .008 | .008 | .008 | .003 | .003 | .006 | .002 | .003 | .005 | .003 | .002 | .001 | .002 |
| **GW Graphon 4** | .151 | .060 | .037 | .018 | .017 | .014 | .018 | .018 | .013 | .011 | .012 | .010 | .010 | .010 | .010 |
| **Std Graphon 4** | .009 | .017 | .012 | .005 | .005 | .002 | .004 | .006 | .003 | .001 | .002 | .002 | .001 | .002 | .001 |

### 5.2 MomentMixup Evaluation

To evaluate the performance of our **MomentMixup** framework, we conducted graph classification experiments on several real-world datasets: AIDS [27], IMDB-Binary [39], IMDB-Multi [39], and Reddit-Binary [39]. Detailed descriptions of these datasets are provided in Appendix K. To ensure a fair comparison with prior work, we adopted the same data splitting methodology as reported in previous literature [3, 13]. For data augmentation, we treated $\alpha_{mix}$, $N_{nodes}$, $N_{graph}$, and $N_{sample}$ as hyperparameters in Algorithm 1 and the best parameters are provided in Appendix K. We employ the GIN architecture [38] as the graph classification model.

Table 2 presents the model's accuracy on the test set. The results demonstrate that our method achieves a better performance gain over the standard G-Mixup approach on three datasets. As highlighted in the previous section, our method demonstrates a distinct advantage on datasets composed of smaller

graphs, such as AIDS, where it notably outperforms techniques like SIGL. While our results on the Reddit-Binary dataset, which features very large graphs and where SIGL performs strongly, were influenced by the experimental choice of using a limited set of nine motifs, this contrast further illuminates a key insight: the optimal choice of mixup method can be highly dependent on graph characteristics, particularly size. Our approach appears particularly well-suited for capturing structural nuances in smaller graphs where fewer motifs can still provide rich representative information.

Table 2: Classification accuracy of G-Mixup, MomentMixup, and baselines on different datasets.

| Dataset | IMDB-B | IMDB-M | REDD-B | AIDS |
|---|---|---|---|---|
| #graphs | 1000 | 1500 | 2000 | 2000 |
| #classes | 2 | 3 | 2 | 2 |
| #avg.nodes | 19.77 | 13.00 | 429.63 | 15.69 |
| #avg.edges | 96.53 | 65.94 | 497.75 | 16.2 |
| GIN | | | | |
| No Augmentation | $71.55\pm3.53$ | $48.83\pm2.75$ | $91.78\pm1.09$ | $98\pm1.2$ |
| G-Mixup w/ USVT | $71.94\pm3.00$ | $50.46\pm1.49$ | $91.32\pm1.51$ | $97.8\pm0.9$ |
| G-Mixup w/ SIGL | $73.95\pm2.64$ | $50.70\pm1.41$ | $\mathbf{92.25}\pm1.41$ | $97.3\pm1$ |
| MomentMixup | $\mathbf{74.30}\pm2.70$ | $\mathbf{50.95}\pm1.93$ | $91.8\pm1.2$ | $\mathbf{98.5}\pm0.6$ |

## 6 Conclusions

In this paper, we introduced a novel, scalable graphon estimator leveraging INRs via direct moment matching, called MomentNet. This approach bypasses latent variables and costly GW optimizations, offering a theoretically grounded, polynomial-time solution for estimating graphons from empirical subgraph counts, with provable guarantees on its accuracy. We further proposed MomentMixup, a new data augmentation technique that performs mixup in the moment space, then obtains the estimated graphon using MomentNet, and finally samples new graphs from this graphon. Our empirical results validate the effectiveness of our estimator, demonstrating superior or comparable performance against state-of-the-art methods in graphon estimation benchmarks, and show that MomentMixup improves graph classification accuracy by generating structurally meaningful augmented data.

Despite its strengths, our method's reliance on a pre-selected set of moments for graphon estimation is a limitation; performance can degrade if these moments are insufficient or noisy. Additionally, modeling a single graphon (per class for MomentMixup) may not capture highly heterogeneous graph data. Future work could address these by developing adaptive moment selection techniques and exploring extensions to learn mixtures of graphons. Further enhancements include adapting our moment-based approach for attributed or dynamic networks and integrating feature learning into the estimation process, broadening the applicability of our framework.

## 7 Acknowledgments

This work was partially supported by the U.S. NSF under award CCF-2340481, the Spanish AEI (AEI/10.13039/501100011033) grant PID2023-149457OB-I00, and the Community of Madrid via grants IDEA-CM (TEC-2024/COM-89) and URJC/CAM F1180 and the ELLIS Madrid Unit.

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

# A Detailed Related Work

**Graphon Estimation**   Graphon estimation aims to recover the underlying generative structure of observed networks. Classical approaches include methods based on histogram estimators by partitioning nodes according to degree or other structural properties [5, 18, 6], and fitting stochastic block models (SBMs) or their variants, which can be viewed as piecewise constant graphon estimators [1, 11]. Universal singular value thresholding (USVT) [7] offers a non-parametric approach for estimating graphons from a single adjacency matrix, particularly effective for low-rank structures. However, many of these methods face challenges in terms of computational cost for large graphs, achieving resolution-free approximation, or may rely on specific structural assumptions (e.g., piecewise constant for SBMs).

More recently, scalable graphon estimation techniques have gained prominence. For example, some works aim at minimizing distances between graph representations but often involve computationally expensive metrics like the GW distance [24, 37, 35], which can be a bottleneck for large networks. The advent of INRs has opened new avenues for continuous, resolution-free graphon estimation. For instance, IGNR (Implicit Graphon Neural Representation) [34] proposed to directly model graphons using neural networks, enabling the representation of graphons up to arbitrary resolutions and efficient generation of arbitrary-sized graphs. IGNR also addresses unaligned input graphs of different sizes by incorporating the Gromov-Wasserstein distance in its learning framework, often within an auto-encoder setup for graphon learning. Subsequently, SIGL (Scalable Implicit Graphon Learning) [3] further advanced INR-based graphon estimation by combining INRs with Graph Neural Networks (GNNs). In SIGL, GNNs are utilized to improve graph alignment by determining appropriate node orderings, aiming to enhance scalability and learn a continuous graphon at arbitrary resolutions, with theoretical results supporting the consistency of its estimator. While these INR-based techniques offer significant advantages in terms of resolution-free representation and handling unaligned data, they still implicitly involve latent variable modeling or rely on GW-like objectives for alignment. Our proposed method builds upon the representational power of INRs but distinguishes itself by directly recovering the graphon via moment matching. This avoids the need for latent variables, complex metric computations like GW, and provides a theoretically grounded estimation framework that naturally handles multiple observed graphs by matching aggregated empirical moments.

**Data Augmentation for Graph Classification**   Data augmentation is crucial for improving the generalization of GNNs and other graph learning models, especially when labeled data is scarce. Mixup [40], which creates synthetic examples by linearly interpolating pairs of samples and their labels, has shown remarkable success in various domains. Its adaptation to graph data has been explored through several avenues, addressing challenges such as varying node counts, lack of alignment, and the non-Euclidean nature of graphs. For instance, Wang et al. [33] proposed interpolating hidden states of GNNs. Particularly relevant to our work are G-Mixup [13] and GraphMAD Navarro and Segarra [21], which recognize the difficulties of direct graph interpolation and propose to augment graphs for graph classification by operating in the space of graphons. GraphMAD Navarro and Segarra [21] projects graphs into the latent space of graphons and implements nonlinear mixup strategies like convex clustering. G-Mixup [13] first estimates a graphon for each class of graphs from the training data. Then, instead of directly manipulating discrete graph structures, G-Mixup interpolates these estimated graphons of different classes in their continuous, Euclidean representation to obtain mixed graphons. Synthetic graphs for augmentation are subsequently generated by sampling from these mixed graphons. This technique has also been adopted as an augmentation strategy in the evaluation pipelines of some graphon estimation studies for downstream tasks [3].

# B Proof of Theorem 1

## B.1 Supporting Lemmas

We rely on the following established and derived results. Lemma 1 is an original contribution of this work, while Lemma 2 is Theorem 3.7 (b) in Borgs et al. [5] and it is included here for completeness.

**Lemma 1** (Concentration of Empirical Motifs). *Let $F$ be a simple graph with $k = |\mathcal{V}_F|$ vertices. For $P \geq 1$ graphs $G_1, \ldots, G_P$, each sampled independently from $G_n(W^*)$, and for any error tolerance $\epsilon_s > 0$, the probability that the empirical motif density $\bar{t}(F, W^*) = \frac{1}{P} \sum_{p=1}^{P} t(F, G_p)$ deviates from*

*the true motif density $t(F, W^*)$ is bounded as*

$$\mathbb{P}[|\bar{t}(F, W^*) - t(F, W^*)| \geq \epsilon_s] \leq 2 \exp\left(-\frac{Pn}{4k^2}\left(\epsilon_s - \frac{k(k-1)}{2n}\right)^2\right), \qquad (9)$$

*for $\epsilon_s > \frac{k(k-1)}{2n}$.*

*Proof.* Let $X_p = t(F, G_p)$ for $p = 1, \ldots, P$. The graphs $G_p$ are independent samples from $G_n(W^*)$, so the random variables $X_p$ are independent and identically distributed.

We leverage concentration properties of $t(F, G_n(W^*))$ in Borgs et al. [5, Lemma 4.4], stating that $t(F, G_n(W^*))$ is concentrated around $t(F, W^*)$ with probability

$$\mathbb{P}[|t(F, G_n(W^*)) - t(F, W^*)| > \delta] \leq 2 \exp(-n\delta^2/(4k^2)). \qquad (10)$$

This implies that the variable $Z = t(F, G_n(W^*)) - t(F, W^*)$ behaves like a sub-Gaussian random variable [32][2]. Comparing the exponent $-\frac{n\delta^2}{4k^2}$ from (10) with the sub-Gaussian tail exponent $-\frac{\delta^2}{2\sigma^2}$, we see that $t(F, G_n(W^*)) - t(F, W^*)$ is sub-Gaussian with parameter $\sigma_Z^2 = \frac{2k^2}{n}$.

The variables we are averaging are $X_p = t(F, G_p)$ with $G_p \sim G_n(W^*)$. Let $\mu_n = \mathbb{E}[X_p] = \mathbb{E}[t(F, G_n(W^*))]$. The centered variables fulfill $X_p - \mu_n = (t(F, G_p) - t(F, W^*)) - (\mathbb{E}[t(F, G_p)] - t(F, W^*))$. Subtracting a constant (the bias $\mathbb{E}[t(F, G_p)] - t(F, W^*)$) from a sub-Gaussian variable preserves its sub-Gaussian property with the same parameter. Thus, $X_p - \mu_n$ are independent, zero-mean, and $\sigma^2$-sub-Gaussian with $\sigma^2 = \sigma_Z^2 = \frac{2k^2}{n}$.

The average of $P$ independent $\sigma^2$-sub-Gaussian random variables is $(\sigma^2/P)$-sub-Gaussian [32]. Let $\bar{Y} = \frac{1}{P}\sum_{p=1}^{P}(X_p - \mu_n) = \bar{t}(F, W^*) - \mu_n$. Then $\bar{Y}$ is $\left(\frac{2k^2}{nP}\right)$-sub-Gaussian. The tail bound for $\bar{Y}$ is

$$\mathbb{P}[|\bar{Y}| \geq \delta] \leq 2 \exp\left(-\frac{\delta^2}{2 \cdot \frac{2k^2}{nP}}\right) = 2 \exp\left(-\frac{\delta^2 nP}{4k^2}\right). \qquad (11)$$

Substituting $\bar{Y} = \bar{t}(F, W^*) - \mu_n$, we get the concentration bound for the empirical mean around the expected mean:

$$\mathbb{P}[|\bar{t}(F, W^*) - \mu_n| \geq \delta] \leq 2 \exp\left(-\frac{\delta^2 nP}{4k^2}\right). \qquad (12)$$

We are interested in the deviation of $\bar{t}(F, W^*)$ from the true motif density $t(F, W^*)$. We use the triangle inequality to relate this deviation to the deviation from the mean $\mu_n$

$$|\bar{t}(F, W^*) - t(F, W^*)| \leq |\bar{t}(F, W^*) - \mu_n| + |\mu_n - t(F, W^*)|. \qquad (13)$$

Let $B_n = |\mu_n - t(F, W^*)|$ be the bias of the empirical estimate. It is known from the theory of graph limits (e.g., related to Borgs et al. [5, Lemma 4.3]) that this bias is bounded by $B_n \leq \frac{k(k-1)}{2n}$. If the deviation from the true density is at least $\epsilon_s$, i.e., $|\bar{t}(F, W^*) - t(F, W)| \geq \epsilon_s$, then it must be that $|\bar{t}(F, W^*) - \mu_n| \geq \epsilon_s - B_n$. This implication requires $\epsilon_s > B_n$ for the bound to be meaningful. Thus, for $\epsilon_s > B_n$

$$\mathbb{P}[|\bar{t}(F, W^*) - t(F, W^*)| \geq \epsilon_s] \leq \mathbb{P}[|\bar{t}(F, W^*) - \mu_n| \geq \epsilon_s - B_n]. \qquad (14)$$

Using the inequality (12) with $\delta = \epsilon_s - B_n$

$$\mathbb{P}[|\bar{t}(F, W^*) - t(F, W^*)| \geq \epsilon_s] \leq 2 \exp\left(-\frac{(\epsilon_s - B_n)^2 nP}{4k^2}\right). \qquad (15)$$

Introducing the upper bound for the bias, $B_n \leq \frac{k(k-1)}{2n}$

$$\mathbb{P}[|\bar{t}(F, W^*) - t(F, W^*)| \geq \epsilon_s] \leq 2 \exp\left(-\frac{\left(\epsilon_s - \frac{k(k-1)}{2n}\right)^2 nP}{4k^2}\right) \qquad (16)$$

$$= 2 \exp\left(-\frac{Pn}{4k^2}\left(\epsilon_s - \frac{k(k-1)}{2n}\right)^2\right). \qquad (17)$$

---

[2] A random variable $Y$ is $\sigma^2$-sub-Gaussian if $\mathbb{E}[e^{\lambda Y}] \leq e^{\lambda^2\sigma^2/2}$ for all $\lambda \in \mathbb{R}$, which implies the tail bound $\mathbb{P}[|Y| \geq \delta] \leq 2e^{-\delta^2/(2\sigma^2)}$.

This bound is valid when $\epsilon_s > \frac{k(k-1)}{2n}$, as required by the lemma statement. $\qquad\square$

**Lemma 2** (Motif Proximity Implies Cut Distance Proximity (Borgs et al. [5], Theorem 3.7 (b)))**.**
*For any integer $k \geq 1$, if the motif distance between two graphons $W_1$ and $W_2$ fulfills $|t(F, W_1) - t(F, W_2)| < \delta_M = 3^{-k^2}$ for every simple graph $F \in \mathcal{F}_k$, then the cut distance between $W_1$ and $W_2$ is upper bounded by*

$$d_{cut}(W_1, W_2) \leq \eta = \frac{22C}{\sqrt{\log_2 k}}, \tag{18}$$

*where $C = \max\{1, \|W_1\|_\infty, \|W_2\|_\infty\}$.*

## B.2 A comment on Assumption 1

This assumption is fundamentally supported by the Universal Approximation Theorem (UAT) [8, 14, 16]. The UAT posits that a neural network with sufficient capacity (e.g., an adequate number of neurons in one or more hidden layers and appropriate non-linear activation functions) can approximate any continuous function to an arbitrary degree of accuracy on a compact domain. In our context, the INR $f_\theta$ models the graphon $W : [0,1]^2 \to [0,1]$. The motif density $t(F, W)$ (as defined in Equation 1) is a continuous functional of $W$, meaning small changes in $W$ lead to small changes in $t(F, W)$. Consequently, if the INR $f_\theta$ can approximate any continuous graphon function, it can learn a specific $f_\theta$ such that the motif densities of the graphon estimated by the INR $t(F, f_\theta)$ are arbitrarily close to some target values. Given that our estimated motif densities $\hat{t}_\theta(F, \hat{W}_\theta)$ are Monte Carlo approximations of $t(F, f_\theta)$, they too can approach these target values (the empirical densities $\bar{t}(F, W^*)$) as the approximation of the underlying function by $f_\theta$ improves as the number of Monte Carlo samples $L$ increases. The assumption thus relies on the INR's capacity to learn a suitable graphon function $f_\theta$ and the optimization process's ability to find the parameters $\theta$ that make the resulting motif estimates $\hat{t}_\theta(F, \hat{W}_\theta)$ match the empirical observations $\bar{t}(F, W^*)$.

## B.3 Proof of Theorem 1

*Proof of Theorem 1.* Our goal is to bound the cut distance $d_{\text{cut}}(\hat{W}_\theta, W^*)$ by $\eta$, which is achieved if we can show that $|\hat{t}(F, \hat{W}_\theta) - t(F, W^*)| < \delta_M$ for all simple graphs $F$ with $|\mathcal{V}_F| = k$ and where the values of both $\eta$ and $\delta_M$ are provided in Lemma 2.

Consider any graph $F \in \mathcal{F}_k$. Using the triangle inequality, we can bound the difference between the neural network's motif estimate and the true graphon motif

$$|\hat{t}_\theta(F, \hat{W}_\theta) - t(F, W^*)| \leq |\hat{t}_\theta(F, \hat{W}_\theta) - \bar{t}(F, W^*)| + |\bar{t}(F, W^*) - t(F, W^*)|. \tag{19}$$

By Assumption 1 on the neural network's training performance, we guarantee

$$|\hat{t}_\theta(F, \hat{W}_\theta) - \bar{t}(F, W^*)| < \epsilon_a = \frac{\delta_M}{2}, \tag{20}$$

for every $F \in \mathcal{F}_k$.

Now we need to bound the second term in the right-hand side of (19), the deviation of the empirical motif density from the true motif density $|\bar{t}(F, W^*) - t(F, W^*)|$. We use Lemma 1 with the sampling error tolerance set to $\epsilon_s = \frac{\delta_M}{2}$. For this lemma to apply, we require $\epsilon_s > \frac{k(k-1)}{2n}$, which is equivalent to $\frac{\delta_M}{2} > \frac{k(k-1)}{2n}$, or $n > \frac{k(k-1)}{\delta_M}$. This condition is enforced in the theorem statement.

For a *specific* graph $F \in \mathcal{F}_k$, the probability that the sampling error is large is bounded by Lemma 1

$$\mathbb{P}\left[|\bar{t}(F, W^*) - t(F, W^*)| \geq \frac{\delta_M}{2}\right] \leq 2\exp\left(-\frac{Pn}{4k^2}\left(\frac{\delta_M}{2} - \frac{k(k-1)}{2n}\right)^2\right). \tag{21}$$

Let $\mathbb{P}_{\text{fail}, F}$ denote this upper bound for a single graph $F \in \mathcal{F}_k$. However, we require the sampling error $|\bar{t}(F, W^*) - t(F, W^*)|$ to be less than $\frac{\delta_M}{2}$ for *all* graphs $F \in \mathcal{F}_k$ simultaneously. By the union bound, the probability that there exists at least one graph $F \in \mathcal{F}_k$ for which the sampling error is $\frac{\delta_M}{2}$ or more is at most the sum of the probabilities for each individual graph

$$\mathbb{P}[\exists F \in \mathcal{F}_k \text{ s.t. } |\bar{t}(F, W^*) - t(F, W^*)| \geq \frac{\delta_M}{2}] \leq \sum_{F \in \mathcal{F}_k} \mathbb{P}_{\text{fail}, F}. \tag{22}$$

Since $|\mathcal{V}_F| = k$ for all $F \in \mathcal{F}_k$, the bound $\mathbb{P}_{\text{fail},F}$ is identical for all these graphs. The sum is thus $N_k \cdot \mathbb{P}_{\text{fail},F}$, where we recall that $N_k = |\mathcal{F}_k|$. The condition (7) in the theorem is precisely set to ensure that this total probability of failure is less than the desired confidence level $\zeta$

$$N_k \cdot 2 \exp\left(-\frac{Pn}{4k^2}\left(\frac{\delta_M}{2} - \frac{k(k-1)}{2n}\right)^2\right) < \zeta. \tag{23}$$

Therefore, with probability at least $1 - \zeta$ (over the random graph samples $G_p$), the event that $|\bar{t}(F, W^*) - t(F, W^*)| < \frac{\delta_M}{2}$ holds for all $F \in \mathcal{F}_k$ occurs.

Conditioned on this high-probability event, and using the neural network approximation in Assumption 1, we have for every $F \in \mathcal{F}_k$

$$|\hat{t}_\theta(F, \hat{W}_\theta) - t(F, W^*)| \leq |\hat{t}_\theta(F, \hat{W}_\theta) - \bar{t}(F, W^*)| + |\bar{t}(F, W^*) - t(F, W^*)| < \frac{\delta_M}{2} + \frac{\delta_M}{2} = \delta_M. \tag{24}$$

Since $|\hat{t}_\theta(F, \hat{W}_\theta) - t(F, W^*)| < \delta_M$ holds for all $F \in \mathcal{F}_k$, Lemma 2 implies that the cut distance between the estimated graphon $\hat{W}_\theta$ and the true graphon $W^*$ is less than $\eta$

$$d_{\text{cut}}(W_\theta, W) < \eta, \tag{25}$$

with probability at least $1 - \zeta$, concluding the proof. $\square$

## C  Equation (7) Bound's Applicability in Realistic Regimes

Assuming a small motif error $\delta_M$ is achievable, we now show that the overall probabilistic bound from (7) is non-vacuous for realistic dataset sizes. Table 3 shows the minimum value of $\zeta$ (left-hand side of equation (7)) for $k = 4$, a conservatively large motif error of $\delta_M = 0.07$ (which is orders of magnitude larger than our empirical results) and various numbers of graphs $P$ and nodes $n$.

Table 3: Minimum value of $\zeta$ in equation (7) for varying values of $P$ and $n$.

| $n$ \ $P$ | 400 | 600 | 800 | 1000 | 1200 | 1400 | 1600 | 1800 | 2000 |
|---|---|---|---|---|---|---|---|---|---|
| 200 | 11.63 | 11.45 | 11.27 | 11.10 | 10.93 | 10.76 | 10.59 | 10.43 | 10.26 |
| 250 | 9.93 | 9.04 | 8.22 | 7.48 | 6.81 | 6.19 | 5.63 | 5.13 | 4.66 |
| 300 | 7.87 | 6.37 | 5.16 | 4.18 | 3.39 | 2.74 | 2.22 | 1.80 | 1.46 |
| 350 | 5.97 | 4.22 | 2.97 | 2.10 | 1.48 | 1.04 | 0.74 | 0.52 | 0.37 |
| 400 | 4.42 | 2.68 | 1.62 | 0.99 | 0.60 | 0.36 | 0.22 | 0.13 | 0.08 |
| 450 | 3.21 | 1.66 | 0.86 | 0.44 | 0.23 | 0.12 | 0.06 | 0.03 | 0.016 |

Although values $> 1$ are uninformative as probabilities, the table clearly shows the bound's exponential decay. Crucially, the guarantee becomes meaningful for realistic data regimes. For example, the REDDIT-B dataset has $P = 2000$ graphs with an average $n \approx 497$. Our analysis shows that in a comparable setting ($n = 450$, $P = 2000$), the failure probability $\zeta$ is a practically useful 0.016. This confirms that the theoretical conditions for our guarantee to hold are met within realistic data regimes.

## D  Unbiasedness of Monte Carlo Estimator for an INR-Based Graphon Moment Estimator

Let $F = (\mathcal{V}_F, \mathcal{E}_F)$ be a graph, where $\mathcal{V}_F$ is a set of $k = |\mathcal{V}_F|$ vertices and $\mathcal{E}_F$ is the set of edges. Let $f_\theta : [0,1]^2 \rightarrow [0,1]$ be an INR parameterized by $\theta$, which models the probability of an edge existing between two nodes based on their latent variables $\eta_i, \eta_j \in [0,1]$, and its estimated graphon is denoted by $\hat{W}_\theta$.

The likelihood of observing the graph structure $F$ given a specific set of latent variable assignments $\boldsymbol{\eta} = \{\eta_v\}_{v \in \mathcal{V}_F}$ and the INR model $f_\theta$ is given by

$$P_\theta(\boldsymbol{\eta}; F, \hat{W}_\theta) = \prod_{(i,j)\in\mathcal{E}_F} \hat{W}_\theta(\eta_i, \eta_j) \prod_{(i,j)\notin\mathcal{E}_F} (1 - \hat{W}_\theta(\eta_i, \eta_j)). \tag{26}$$

The quantity $t'_\theta(F, \hat{W}_\theta)$ is defined as this likelihood integrated over all possible configurations of the latent variables in the $k$-dimensional unit hypercube

$$t'_\theta(F, \hat{W}_\theta) = \int_{[0,1]^k} P_\theta(\boldsymbol{\eta}; F, \hat{W}_\theta)d\boldsymbol{\eta}, \tag{27}$$

where $d\boldsymbol{\eta} = \prod_{v \in \mathcal{V}_F} d\eta_v$.

The $L$-sample Monte Carlo estimator for $t'_\theta(F, \hat{W}_\theta)$ is given by

$$\hat{t}'_\theta(F, \hat{W}_\theta) = \frac{1}{L} \sum_{l=1}^{L} P(\boldsymbol{\eta}^{(l)}; F, \hat{W}_\theta). \tag{28}$$

For this estimation, each sample $\boldsymbol{\eta}^{(l)} = [\eta_{v_1}^{(l)}, \ldots, \eta_{v_k}^{(l)}]$ is a vector where each component $\eta_v^{(l)}$ (for $v \in \mathcal{V}_F$) is drawn independently from the uniform distribution $\mathcal{U}[0, 1]$.

### D.1 Unbiasedness of the Estimator

**Theorem 2.** *The Monte Carlo estimator $\hat{t}'_\theta(F, \hat{W}_\theta)$ is an unbiased estimator of $t'_\theta(F, \hat{W}_\theta)$.*

*Proof.* To show that the Monte Carlo estimation $\hat{t}'_\theta(F, \hat{W}_\theta)$ is an unbiased estimator of $t'_\theta(F, \hat{W}_\theta)$, we need to prove that $\mathbb{E}[\hat{t}'_\theta(F, \hat{W}_\theta)] = t'_\theta(F, \hat{W}_\theta)$.

The expectation of the estimator is:

$$\mathbb{E}[\hat{t}'_\theta(F, \hat{W}_\theta)] = \mathbb{E}\left[\frac{1}{L} \sum_{l=1}^{L} P_\theta(\boldsymbol{\eta}^{(l)}; F, \hat{W}_\theta)\right]$$

$$= \frac{1}{L} \sum_{l=1}^{L} \mathbb{E}[P_\theta(\boldsymbol{\eta}^{(l)}; F, \hat{W}_\theta)] \quad \text{(by linearity of the expectation).} \tag{29}$$

Since each sample $\boldsymbol{\eta}^{(l)}$ is drawn independently from the same uniform distribution, therefore its pdf is $p(\boldsymbol{\eta}) = 1$ on $[0, 1]^k$, the expectation $\mathbb{E}[P_\theta(\boldsymbol{\eta}^{(l)}; F, \hat{W}_\theta)]$ is the same for all $l$. Let this common expectation be $\mathbb{E}[P_\theta(\boldsymbol{\eta}; F, \hat{W}_\theta)]$, whose value is

$$\mathbb{E}[P_\theta(\boldsymbol{\eta}; F, \hat{W}_\theta)] = \int_{[0,1]^k} P_\theta(\boldsymbol{\eta}; F, \hat{W}_\theta)p(\boldsymbol{\eta})d\boldsymbol{\eta}$$

$$= \int_{[0,1]^k} P_\theta(\boldsymbol{\eta}; F, \hat{W}_\theta)) \cdot 1 \, d\boldsymbol{\eta} \quad \text{(since } p(\boldsymbol{\eta}) = 1 \text{ on } [0, 1]^k\text{)}$$

$$= t'_\theta(F, \hat{W}_\theta),$$

according to (27). Substituting this back into (29)

$$\mathbb{E}[\hat{t}'_\theta(F, \hat{W}_\theta)] = \frac{1}{L} \sum_{l=1}^{L} t'_\theta(F, \hat{W}_\theta)$$

$$= \frac{1}{L}(L \cdot t'_\theta(F, \hat{W}_\theta))$$

$$= t'_\theta(F, \hat{W}_\theta).$$

Thus, $\mathbb{E}[\hat{t}'_\theta(F, \hat{W}_\theta)] = t'_\theta(F, \hat{W}_\theta)$, which shows that the Monte Carlo estimator $\hat{t}'_\theta(F, \hat{W}_\theta)$ is an unbiased estimator of $t'_\theta(F, \hat{W}_\theta)$. This means that, on average, the estimator will yield the true value of the integral defined by $f_\theta$ and the graph structure $F$. $\square$

## E  Time Complexity of MOMENTNET

**Stage 1: parallel motif–density extraction.** For each graph $G_p = (\mathcal{V}_p, \mathcal{E}_p)$ let $n_p = |\mathcal{V}_p|$, $e_p = |\mathcal{E}_p|$ and $d_p = \max_{v \in \mathcal{V}_p} \deg(v)$ be the number of nodes, number of edges, and maximum degree of the graph $G_p$. ORCA [15] counts all $2-4$-node graphlets in

$$T_{\text{ORCA}}(G_p) = O\big(e_p d_p + n_p d_p^3\big).$$

Because every graph can be processed independently, we dispatch the $P$ graphs to $M$ workers $(M \leq P)$. Hence the *wall-clock* preprocessing time is

$$T_{\text{stage 1}} = O\Big(\lceil \tfrac{P}{M} \rceil \max_p (e_p d_p + n_p d_p^3)\Big).$$

With one worker per graph $(M = P)$ this shrinks to the single-graph cost that dominates $(\max_p)$.

**Stage 2: training the Moment network.**   Define:

- $L$: number of Monte-Carlo samples per epoch;
- $N_e$: number of training epochs;
- $C_{\text{INR}}$: cost of one forward/back-prop through the INR for a single edge probability;
- $|\theta|$: total number of trainable parameters.

Each motif instance $F$ of size $|\mathcal{V}_F| \leq 4$ invokes the INR at most six times, a constant. One epoch therefore costs

$$T_{\text{epoch}} = O\big(L\,C_{\text{INR}} + |\theta|\big), \qquad T_{\text{stage 2}} = O\big(N_e\,(L\,C_{\text{INR}} + |\theta|)\big).$$

**Overall wall-clock complexity.**

$$T_{\text{MomentNet}} = O\Big(\lceil \tfrac{P}{M} \rceil \max_p (e_p d_p + n_p d_p^3) + N_e\,(L\,C_{\text{INR}} + |\theta|)\Big).$$

**Comparison with SIGL in Sparse vs. Dense Regimes**

SIGL [3] requires message-passing GNN training, histogram building and INR fitting; with $N_e$ epochs its wall-clock cost is $T_{\text{SIGL}} = O(P N_e n_T^2)$, where $n_T = \max_p n_p$.

- **Sparse regime** ($d_{\max} = O(1) \Rightarrow e_p = O(n_p)$):
  - MOMENTNET: $T = O(\lceil \tfrac{P}{M} \rceil n_T + N_e\,(L\,C_{\text{INR}} + |\theta|))$;
  - SIGL: $T = O(P N_e n_T^2)$.

  Here MomentNet grows *linearly* in $n_T$ (plus the network-training term), whereas SIGL is quadratic. In practice we repeatedly observe MomentNet to be faster when graphs have $e_p = O(n_p)$ even for very large $n_p$.

- **Dense regime** (Erdős–Rényi with $p_{conn} = 0.5$ implies $d_{\max} \approx n_T/2$ and $e_p = \Theta(n_T^2)$):
  - MOMENTNET: $T = O(\lceil \tfrac{P}{M} \rceil n_T^4 + N_e\,(L\,C_{\text{INR}} + |\theta|))$;
  - SIGL: $T = O(P N_e n_T^2)$.

  Asymptotically, SIGL's $n_T^2$ term is smaller than MomentNet's $n_T^4$. Yet empirical runs on dense ER graphs with $p_{conn} = 0.5$ still show MomentNet to be faster once (i) Stage 1 is fully parallelised and (ii) the constants behind GNN message passing and histogramming dominate SIGL's quadratic term. Thus, the theoretical advantage of SIGL in dense graphs does not necessarily translate into shorter wall-clock times. Furthermore, MomentNet utilizes a two-stage process. The initial stage involves computing motif counts from the input graphs. Following this, the graphs are discarded. The second stage, which our experiments show to be the dominant phase of our method, then trains an INR using a vector of average moments derived from these counts. This design provides a significant reason for our method's improved speed, particularly in dense scenarios. By isolating the computationally expensive motif counting to a preliminary step, this cost is bypassed during the subsequent, dominant INR learning phase.

With graph-level parallelism, MOMENTNET is *provably linear* in the number of edges for sparse networks and remains competitive on dense networks because its constant factors are smaller and its training cost is independent of the graph size.

## E.1 Justification for Practical Scalability Over SIGL

The theoretical complexity analysis highlights that in the dense regime, MOMENTNET's Stage 1 cost is bounded by $O(\lceil \frac{P}{M} \rceil n_T^4)$, which is asymptotically worse than SIGL's $O(PN_e n_T^2)$. However, as demonstrated in our empirical results (Figure 2c), MOMENTNET is practically faster on large, dense graphs. This is due to two critical factors:

1. **Loose Theoretical Bound and Small Empirical Constant for ORCA:** The worst-case $O(n_T^4)$ bound for ORCA's 4-node motif counting is known to be non-tight. We conducted an empirical analysis by measuring the wall-clock execution time for counting 4-node graphlets on a sequence of dense graphs with node counts $n$ ranging from 30 to 400.

   **Empirical Analysis of ORCA Runtime** We modeled the relationship between runtime $T(n)$ and node count $n$ using a linear regression on log-transformed data, $T(n) \approx c \cdot n^k$. Our results determined the practical growth rate to be nearly cubic, with an exponent of $k \approx 3$ ($R^2 \approx 0.97$). Crucially, the fitted constant factor was extremely small, $c \approx 2.97 \times 10^{-8}$. This empirical finding confirms that the algorithm is highly efficient on dense graphs, ensuring that MOMENTNET's theoretical worst-case cost is practically masked by SIGL's larger constant factors and overheads until $n_T$ becomes very large.

2. **Strategic Subsampling Capability:** Our method's performance relies on the moment vector $\mathbf{m}$, which is robustly estimated by averaging moments from multiple, smaller graphs ($P$ graphs of size $n_p$). MOMENTNET's theoretical guarantee (Theorem 1 and Figure 2b) allows us to strategically subsample a single large input graph into a collection of smaller graphs, effectively reducing the dominant $\max_p n_p$ in Stage 1. This capability lets us trade off $n_p$ for $P$ to minimize runtime while preserving near-optimal performance, a flexibility SIGL lacks. SIGL must process the full-size graph to properly learn latent representations, directly locking its runtime to the high cost of a single, large $n_T$.

Therefore, MOMENTNET's practical speed advantage stems from a combination of a lower-than-worst-case empirical complexity in Stage 1 and a flexible sampling strategy that SIGL cannot utilize.

## E.2 Subsampling Effect on Scalability

The scalability evaluation in Figure 2 demonstrates the conditions under which MomentNet achieves a computational advantage over SIGL. A key insight from our analysis is that SIGL's performance deteriorates significantly on smaller graphs due to its dependence on stable latent variable estimation. This limitation benefits our approach: by *subsampling* a very large graph into a smaller, more manageable collection of subgraphs, we can maintain near-optimal estimation quality while drastically reducing the computational cost. In contrast, running SIGL directly on these smaller subgraphs is ineffective, as its accuracy drops off steeply.

To further validate the benefit of subsampling in the large-graph regime, we constructed a dataset consisting of ten large graphs, each containing 2,000 nodes, sampled from the dense-graph graphon class used in our scalability analysis (Section 5.1.2). We then extracted ten 50-node subgraphs from each 2,000-node graph and used these smaller subgraphs as inputs for both methods. The results, summarized in Table 4, confirm that MomentNet outperforms SIGL in both accuracy (lower GW Loss) and runtime under this efficient subsampling strategy. Crucially, while SIGL's runtime increases sharply with full graph size, our method maintains comparable performance even when utilizing these smaller subgraphs.

Table 4: Performance comparison on large graphs (2,000 nodes) using a subsampling strategy (10 extracted 50-node subgraphs per large graph).

| Method | Training Runtime (s) | GW Loss | Motif Counting Runtime (s) |
|---|---|---|---|
| **MomentNet** | $54.83 \pm 18.69$ | $0.0548 \pm 0.0016$ | $1.59 \pm 0.24$ |
| **SIGL** | $207.89 \pm 18.4$ | $0.1085 \pm 0.0156$ | - |

# F  Proof of Proposition 1

Let $W_1, W_2 \colon [0,1]^2 \to [0,1]$ be two graphons and fix $\alpha \in (0,1)$. Denote their convex combination by

$$W_\alpha = \alpha\, W_1 + (1-\alpha)\, W_2.$$

**Edge density (a linear functional).**  For the single–edge motif $F_e$ on vertices $\mathcal{V}_{F_e} = \{1,2\}$ and $\mathcal{E}_{F_e} = \{(1,2)\}$, the induced density is

$$t'(F_e, W) = \int_{[0,1]^2} W(\eta_1, \eta_2)\, d\eta_1\, d\eta_2 = \mathbb{E}[W(\eta_1, \eta_2)].$$

Because the integrand is *linear* in $W$, we immediately have

$$t'(F_e, W_\alpha) = \alpha\, t'(F_e, W_1) + (1-\alpha)\, t'(F_e, W_2),$$

so the edge density behaves affinely under convex combinations.

**The $V$–shape motif.**  Let $F$ be the $V$–shape (three-vertex path) on vertex set $\mathcal{V}_F = \{1,2,3\}$ and edge set $\mathcal{E}_F = \{(1,2), (1,3)\}$. Its induced density is

$$t'(F, W) = \int_{[0,1]^3} W(\eta_1, \eta_2)\, W(\eta_1, \eta_3) \left[ 1 - W(\eta_2, \eta_3) \right] d\eta_1 d\eta_2 d\eta_3. \tag{30}$$

Plugging $W_\alpha$ into (30)

$$
\begin{aligned}
t'(F, W_\alpha) =& \mathbb{E}\Big[ \big(\alpha W_1 + (1-\alpha)W_2\big)_{12} \big(\alpha W_1 + (1-\alpha)W_2\big)_{13} \big(1 - \alpha W_1 - (1-\alpha)W_2\big)_{23} \Big] \\
=& \alpha^3\, \mathbb{E}\big[(W_1)_{12}(W_1)_{13}(1 - (W_1)_{23})\big] + (1-\alpha)^3\, \mathbb{E}\big[(W_2)_{12}(W_2)_{13}(1 - (W_2)_{23})\big] \\
& + \textbf{mixed terms},
\end{aligned}
\tag{31}
$$

where, to simplify notation, we used $(\cdot)_{ij}$ to denote that the graphon inside the parenthesis is evaluated on $(\eta_i, \eta_j)$ and "mixed terms" contain products in which at least one factor comes from $W_1$ and another from $W_2$. Because these mixed terms generally do not cancel, the right-hand side of (31) *does not* reduce to the affine combination

$$\alpha\, t'(F, W_1) + (1-\alpha)\, t'(F, W_2), \tag{32}$$

except in degenerate cases (e.g. $W_1 = W_2$ or $\alpha \in \{0,1\}$).

**Concrete counter-example.**  Take constant graphons $W_1(\eta_i, \eta_j) = p_1$ and $W_2(\eta_i, \eta_j) = p_2$ with $p_1$ and $p_2$ being constants satisfying $0 < p_1 \neq p_2 < 1$. Then $W_\alpha(\eta_i, \eta_j) = p_\alpha = \alpha p_1 + (1-\alpha) p_2$, and

$$t'(F, W_i) = p_i^2 (1 - p_i), \qquad t'(F, W_\alpha) = p_\alpha^2 (1 - p_\alpha),$$

for $i \in \{1,2\}$. However,

$$p_\alpha^2 (1 - p_\alpha) \neq \alpha p_1^2 (1 - p_1) + (1-\alpha) p_2^2 (1 - p_2)$$

whenever $p_1 \neq p_2$ and $\alpha \in (0,1)$, confirming that the $V$–shape moment is *not* affine in $W$.

**Conclusion.**  Edge moments are linear in the graphon, but higher-order induced moments involve *non-linear* (polynomial) combinations of $W$. Consequently, a convex combination of graphons preserves edge moments but fails to preserve the remaining components of the motif-moment vector. $\qquad\square$

# G  Methods Details

## G.1  Latent Variable Invariance of MomentNet

The graphon model and our proposed model to learn it exhibit invariance to the specific ordering or labeling of latent variables. This means that the estimated graphon is unchanged under measure

preserving transformations [5]. In other words, if the underlying structure of a graphon is rearranged or relabeled, MomentNet can still accurately capture the essential underlying connectivity patterns. To illustrate this crucial property, we conduct an experiment using an SBM graphon, more precisely the one indexed by 12 in Table 7. For this experiment, we utilize the same dataset that was generated for the performance comparison of MomentNet discussed in Section 5. The learned graphons for three different realizations of this experiment are presented in Figure 3. It is evident that all three estimated graphons closely resemble the ground truth graphon, which is depicted in Figure 4. Also, the three estimated graphons reflect the same underlying structure, and all of them share a similar GW loss, which is a loss function invariant to measure preserving transformations. This essentially means that, no matter which of the three depicted graphons we sample graphs from, the underlying structure of all these graphs will be the same. This outcome strongly verifies that MomentNet's primary mechanism involves matching the moments of the graph, without caring about the ordering of the latent variables. Consequently, and in contrast to other methodologies, its estimated graphon accurately reflects the ground truth structure, allowing for differences only up to a permutation of the latent variable locations.

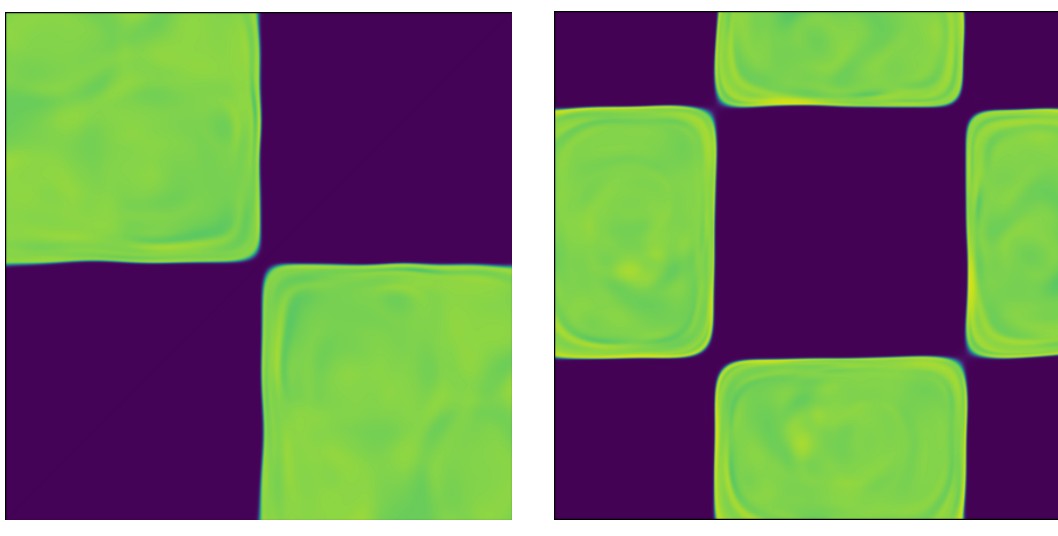

(a) Estimated SBM graphon (Sample 1).          (b) Estimated SBM graphon (Sample 2).

(c) Estimated SBM graphon (Sample 3).

Figure 3: Three samples of estimated graphons derived from a SBM.

## G.2 MomentNet Generalization

To test the generalization capabilities of MomentNet beyond standard metrics like the GW loss and centrality measures, we conducted an additional experiment focusing on moment **extrapolation**. Following the experimental setup described in the paper, we trained a model exclusively on the nine motifs corresponding to 2- to 4-node subgraphs ($F_0$ to $F_8$). We then evaluated its ability to accurately predict the densities of a different, unobserved set of subgraphs: the 5-node motifs (motif indices 10 through 30 in the ORCA paper [15]). The relative error for each extrapolated moment is presented in Table 5. The highly accurate estimations, with a median relative error of less than $2\%$, demonstrate that MomentNet successfully learned the true underlying continuous data distribution (the graphon) from the low-order moments, allowing it to accurately extrapolate high-order structural properties.

Table 5: Extrapolation Error: Relative error ($\%$) of MomentNet's estimated moments for unobserved 5-node motifs (Indices 10–30), after training only on 2- to 4-node motifs (Indices 1–9).

| Motif | Rel. Err. (%) | Motif | Rel. Err. (%) | Motif | Rel. Err. (%) | Motif | Rel. Err. (%) | Motif | Rel. Err. (%) | Motif | Rel. Err. (%) |
|---|---|---|---|---|---|---|---|---|---|---|---|
| 10 | 2.224 | 14 | 0.205 | 18 | 0.154 | 22 | 1.102 | 26 | 3.087 | 30 | 11.086 |
| 11 | 1.536 | 15 | 1.031 | 19 | 0.371 | 23 | 2.468 | 27 | 1.753 | | |
| 12 | 1.785 | 16 | 1.899 | 20 | 1.744 | 24 | 0.140 | 28 | 2.062 | | |
| 13 | 0.073 | 17 | 0.462 | 21 | 1.502 | 25 | 1.043 | 29 | 4.010 | | |

## G.3 Monte Carlo Sampling Variance

Based on the relatively high quality of the results achieved by MomentNet, we hypothesize that stable convergence during training is contingent upon a sufficiently low variance in the Monte Carlo estimation of the motif densities (as described in the moment estimator paragraph of Section 3.1, Equation 4). To investigate this hypothesis directly, we designed an experiment to measure the gradient variance as a function of the number of Monte Carlo samples, $L$. For this test, the parameters ($\theta$) of the INR model were held constant while we estimated the gradient's standard deviation over 1,000 runs for varying numbers of samples, $L$. The results, summarized in Table 6, show a clear inverse relationship: the standard deviation consistently decays as $L$ increases. This finding supports our hypothesis and underscores the importance of using a sufficient number of samples, $L$, to ensure stable and efficient optimization.

Table 6: Gradient Stability vs. Monte Carlo Samples ($L$). The standard deviation of the gradient decreases consistently with the number of Monte Carlo samples, supporting the need for a sufficiently large $L$ for stable optimization.

| Monte Carlo Samples ($L$) | Mean Gradient | Std Dev Gradient |
|---|---|---|
| 100 | 0.013005 | 0.004484 |
| 500 | 0.013107 | 0.002031 |
| 1000 | 0.013030 | 0.001387 |
| 5000 | 0.013019 | 0.000647 |
| 10000 | 0.013052 | 0.000439 |
| 20000 | 0.013009 | 0.000325 |
| 50000 | 0.013013 | 0.000193 |
| 100000 | 0.013019 | 0.000134 |

### G.4 MomentMixup Pseudocode

---

**Algorithm 1** MomentMixup Augmentation

---

**Input:** $\alpha_{\text{mix}}$: float, mixing coefficient ($0 \leq \alpha_{\text{mix}} \leq 1$).

      $\mathcal{G}_i, \mathcal{G}_j$: list of graphs, graph datasets for classes $i$ and $j$.

      $y_i, y_j$: integer, label for classes $i$ and $j$.

      $N_{\text{sample}}$: integer, number of graphs to sample from each class dataset to compute average moments.

      $N_{\text{nodes}}$: integer, number of nodes for each new graph.

      $N_{\text{graphs}}$: integer, number of augmented graphs to generate.

**Output:** $\mathcal{G}_{\text{aug}}$: list of graphs and labels, newly generated augmented graphs.

  1:                                                     ▷ Compute average moment vector for class $i$

  2:  $\mathcal{S}_i \leftarrow$ Randomly select $N_{\text{sample}}$ graphs from $\mathcal{G}_i$

  3:  $\mathbf{m}_i \leftarrow \frac{1}{N_{\text{sample}}} \sum_{G \in \mathcal{S}_i} \text{ComputeGraphMoments}(G)$

  4:                                                    ▷ Compute average moment vector for class $j$

  5:  $\mathcal{S}_j \leftarrow$ Randomly select $N_{\text{sample}}$ graphs from $\mathcal{G}_j$

  6:  $\mathbf{m}_j \leftarrow \frac{1}{N_{\text{sample}}} \sum_{G \in \mathcal{S}_j} \text{ComputeGraphMoments}(G)$

  7:  $\mathbf{m}_{\text{target}} \leftarrow \alpha_{\text{mix}} \cdot \mathbf{m}_i + (1 - \alpha_{\text{mix}}) \cdot \mathbf{m}_j$              ▷ Compute target mixed moments

  8:  $y_{\text{target}} \leftarrow \alpha_{\text{mix}} \cdot y_i + (1 - \alpha_{\text{mix}}) \cdot y_j$             ▷ Compute the label for the new samples

  9:  $W_{\text{aug}} \leftarrow \text{MomentNet}(\mathbf{m}_{\text{target}})$                     ▷ Trains MomentNet for $\mathbf{m}_{\text{target}}$

10:  $\mathcal{G}_{\text{aug}} \leftarrow []$                                     ▷ Initialize list for augmented samples

11: **for** $k \leftarrow 1$ to $N_{\text{graphs}}$ **do**

12:     $G_{\text{new}} \leftarrow \text{SampleGraph}(W_{\text{aug}}, N_{\text{nodes}})$                        ▷ Sample new graph

13:     Add $(G_{\text{new}}, y_{\text{target}})$ to $\mathcal{G}_{\text{aug}}$

14: **end for**

15: **return** $\mathcal{G}_{\text{aug}}$

---

## H  List of Graphons

Table 7: Table of Graphons

|   | $W(x,y)$ |
|---|---|
| 1 | $xy$ |
| 2 | $e^{(-(x^{0.7}+y^{0.7}))}$ |
| 3 | $\frac{1}{4}(x^2 + y^2 + \sqrt{x} + \sqrt{y})$ |
| 4 | $\frac{1}{2}(x + y)$ |
| 5 | $(1 + e^{(-2(x^2+y^2))})^{-1}$ |
| 6 | $(1 + e^{(-\max\{x,y\}^2 - \min\{x,y\}^4)})^{-1}$ |
| 7 | $e^{(-\max\{x,y\}^{0.75})}$ |
| 8 | $e^{(-\frac{1}{2}(\min\{x,y\}+\sqrt{x}+\sqrt{y}))}$ |
| 9 | $\log(1 + \max\{x, y\})$ |
| 10 | $|x - y|$ |
| 11 | $1 - |x - y|$ |
| 12 | $0.8\mathbf{I}_2 \otimes \mathbb{1}_{[0,\frac{1}{2}]^2}$ |
| 13 | $0.8(1 - \mathbf{I}_2) \otimes \mathbb{1}_{[0,\frac{1}{2}]^2}$ |

The graphons are also visualized in Figure 4.

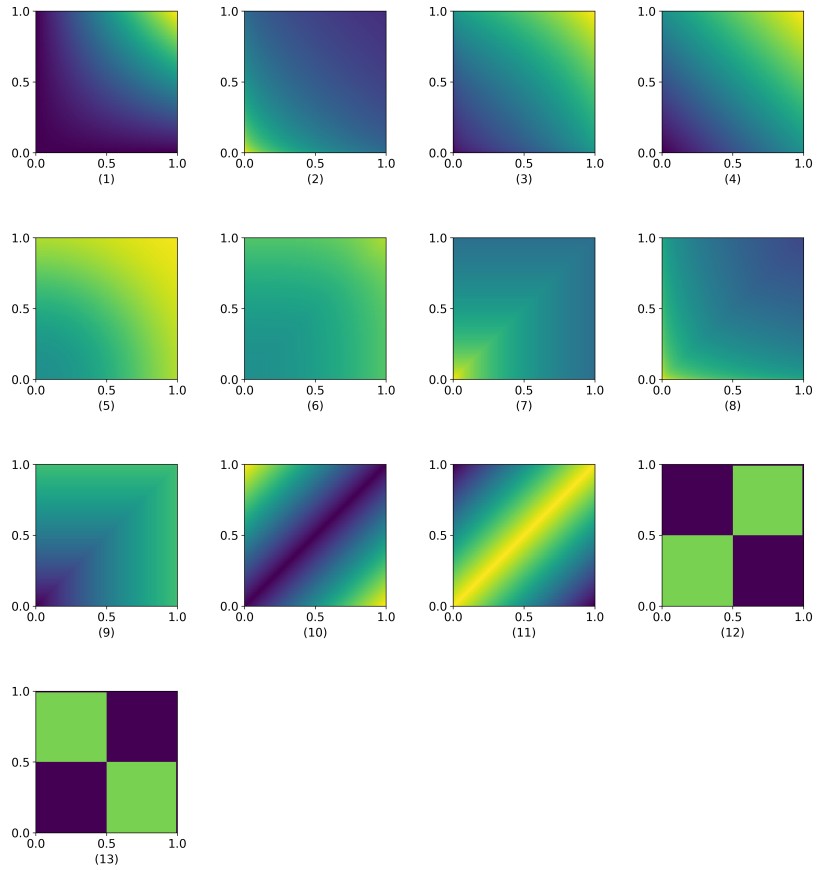

Figure 4: Representation of the graphons defined in Table 7.

# I Selected Motifs

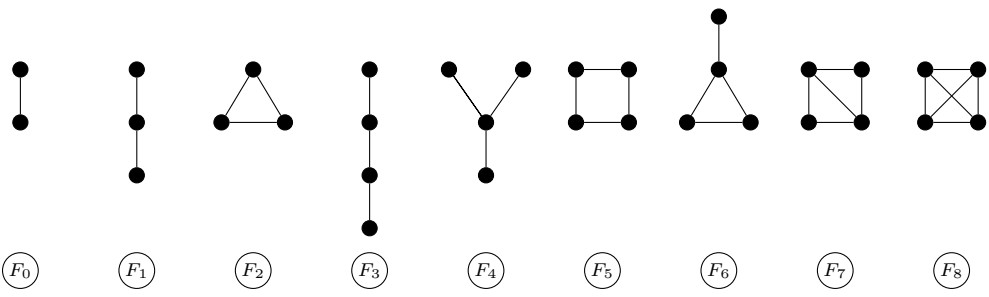

Figure 5: Motifs up to four nodes.

# J Centrality Measures

In real-world graph statistical analysis, **centrality measures** are of significant interest to researchers. Building upon the work of Avella-Medina et al. [2], who demonstrated the computability of these measures on graphons, we use several centrality metrics to further evaluate the quality of the estimated graphons. Specifically, we employ:

- **Degree Centrality**: This measure quantifies the number of direct connections a node possesses.
  - *High Value*: Indicates a node with many direct connections, often acting as a local hub with numerous immediate interactions. Such a node is highly active in its local neighborhood.
  - *Low Value*: Suggests a node with few direct connections, implying less immediate activity or influence within its local vicinity.

- **Eigenvector Centrality**: This identifies influential nodes by considering that connections to other highly-connected (and thus influential) nodes contribute more significantly to a node's score. It measures how well-connected a node is to other well-connected nodes.
  - *High Value*: A node with high eigenvector centrality is connected to other nodes that are themselves influential. This node is likely a key player within an influential cluster or a leader among leaders.
  - *Low Value*: A node with low eigenvector centrality is typically connected to less influential nodes or has relatively few connections overall. Its influence is not strongly amplified by the influence of its neighbors.

- **Katz Centrality**: This measure considers all paths in the graph, assigning exponentially more weight to shorter paths while still accounting for longer ones. It uses an attenuation factor $\alpha$, which determines the weight given to longer paths: smaller values of $\alpha$ emphasize shorter paths, while larger values give more importance to longer paths, up to a theoretical limit to ensure convergence.
  - *High Value*: Indicates a node that is reachable by many other nodes through numerous paths, with shorter paths contributing more. This node is generally well-connected throughout the network, both directly and indirectly, and can efficiently disseminate or receive information.
  - *Low Value*: Suggests a node that is not easily reachable by many other nodes or is primarily connected via very long paths. Its overall influence or accessibility within the network is limited.

- **PageRank Centrality**: Originally developed for web pages, PageRank assesses a node's importance based on the number and quality of its incoming links. A link from an important node carries more weight than a link from a less important one. It uses a damping factor $\beta$, representing the probability that a random walker will follow a link to an adjacent node, while $(1 - \beta)$ is the probability they will jump to a random node in the graph, ensuring that all nodes receive some rank and preventing rank-sinking in disconnected components.
  - *High Value*: A node with high PageRank centrality receives many "votes" (incoming connections) from other important nodes. This indicates that significant entities within the network consider this node to be important or authoritative.
  - *Low Value*: A node with low PageRank centrality receives few incoming connections or is primarily linked by less important nodes. It is not widely recognized as important by other influential nodes in the network.

The mathematical formulations for these graphon-based centrality measures are adopted directly from Avella-Medina et al. [2], corresponding to equations (7), (8), (9), and (10) in their paper, respectively. For a detailed analysis, we focus on graphons 1 and 2, as specified in Table 7. We compute both analytical and sample-based centrality measures, establishing these as baselines for comparison with our results. The analytical computations directly apply the aforementioned formulas from Avella-Medina et al. [2]. For the sample-based approach, we generate discrete graph instances by drawing samples from the ground truth graphon and subsequently compute the centrality measures within this discrete domain. Further details regarding each graphon are presented in the subsequent subsections.

### J.1  Graphon 1: The $(xy)$ Model

The analytical centrality measures formulas for this graphon are as follows:

- **Degree Centrality:**

$$C^d(x) = \frac{x}{2}$$

- **Eigenvector Centrality:**

$$C^e(x) = \sqrt{3}x$$

- **Katz Centrality:**

$$C_\alpha^k(x) = (6 - 2\alpha) + 3\alpha x$$

- **PageRank Centrality:**

$$C_\beta^{\mathrm{pr}}(x) = (1 - \beta) + 2\beta x$$

These measures are for the given latent variable $x \in [0, 1]$, after computing its centrality vector, we normalize it before comparison with discrete graph centralities [2]. Since the ordering for these experiments is important, we create a new dataset of 20 graphs with 100 nodes each, preserving the latent variables for all the nodes. The experiment results are illustrated in Figure 6. Our results show that centrality measures from the MomentNet-predicted graphon (blue lines in the figure) are close to the analytical computations (ground truth, black dashed lines). Furthermore, these graphon-based centralities by MomentNet also provide a good approximation for centrality measures computed over discrete graph samples (red dots).

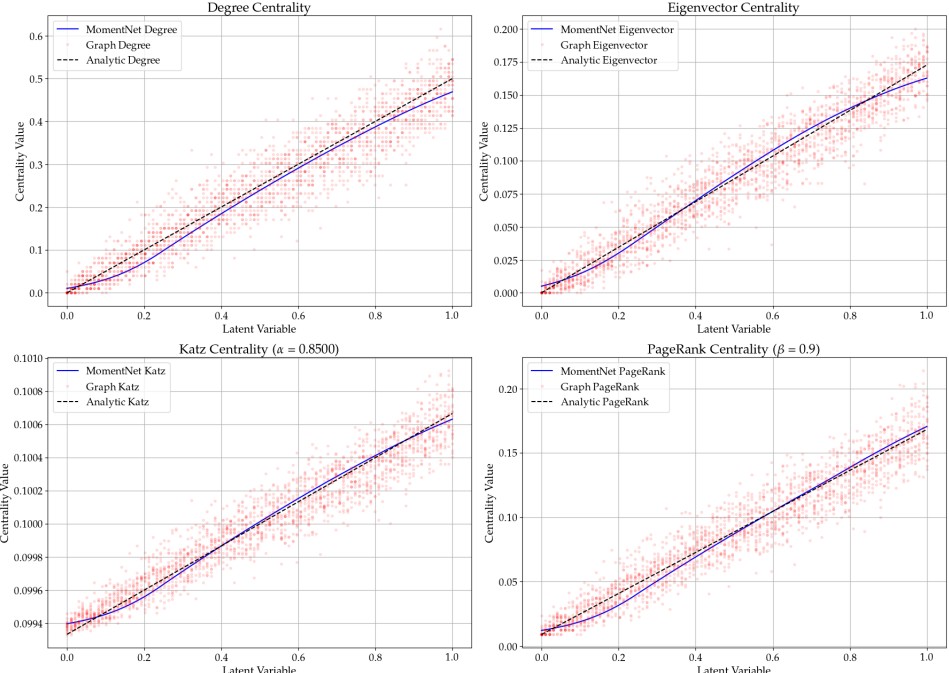

Figure 6: Centrality measures: MomentNet vs. analytic computation for the $xy$ graphon.

## J.2 Graphon 2: The $(e^{(-(x^{0.7}+y^{0.7}))})$ Model

To test the generalizability and consistent performance of our method across varying complexities, we replicated the experiment on a more complex graphon. The analytical centrality measures formulas for this graphon are as follows:

- **Degree Centrality:**

$$C^d(x) = 0.7492 \, e^{-x^{0.7}}$$

- **Eigenvector Centrality:**

$$C^e(x) = \frac{e^{-x^{0.7}}}{\sqrt{0.473}}$$

- **Katz Centrality:**

$$C_\alpha^k(x) = 1 + \frac{0.7492 \, \alpha \, e^{-x^{0.7}}}{1 - 0.473 \, \alpha}$$

- **PageRank Centrality:**

$$C_\beta^{\mathrm{pr}}(x,\beta) = (1-\beta) + \frac{\beta}{0.7492}\, e^{-x^{0.7}}$$

The experiment results are illustrated in Figure 7. Similar to the previous experiment, after computing the centrality measures on the graphon and analytically, we normalize them to compare them with the discrete graph measurement. As the plots show, similar to the previous graphon, our estimation is very close to the ground truth results obtained by analytical calculation.

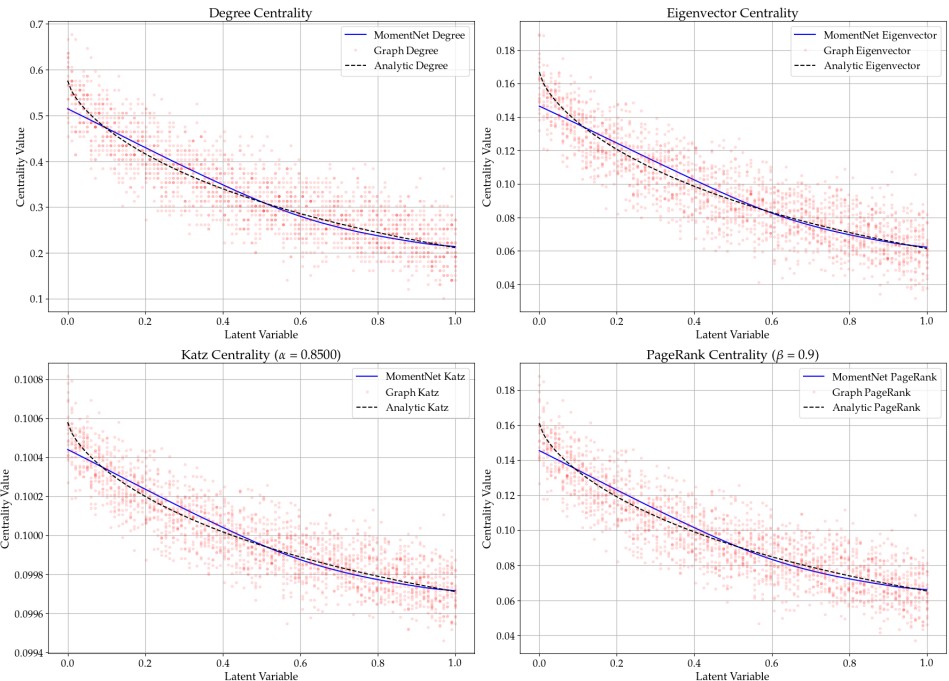

Figure 7: Centrality measures: MomentNet vs. analytic computation for the $e^{(-(x^{0.7}+y^{0.7}))}$ graphon.

## K   Extra Scalability Evaluations

We conducted an additional experiment to evaluate the scalability of SIGL and MomentNet. For this assessment, rather than focusing on SIGL's known weaknesses in latent variable estimation, we selected graphon number 5 from Table 7, a model that both methods accurately estimate. We generate 10 graphs for each node size $n \in \{10, 20, \ldots, 810\}$.

Figure 8 illustrates the scalability of MomentNet and SIGL in terms of both performance, measured by GW loss, and average runtime, as a function of the number of nodes. Subfigure (a) of Figure 8 reveals that MomentNet (blue line) maintains a consistently low GW loss across the tested range of node sizes, indicating stable performance. In contrast, SIGL's (red line) GW loss starts notably higher for smaller networks but decreases substantially as the number of nodes increases, eventually matching or even slightly outperforming MomentNet's loss for larger networks.

However, subfigure (b) of Figure 8 highlights a significant difference in computational efficiency: MomentNet's average runtime exhibits only a modest and gradual increase with the number of nodes. Conversely, SIGL's runtime escalates sharply, demonstrating significantly poorer scalability.

Consequently, while SIGL might offer a marginal advantage in GW Loss for very large graphs, MomentNet's vastly superior runtime scalability makes it a more practical and favorable approach, particularly for applications involving large-scale networks where computational resources and time are critical factors.

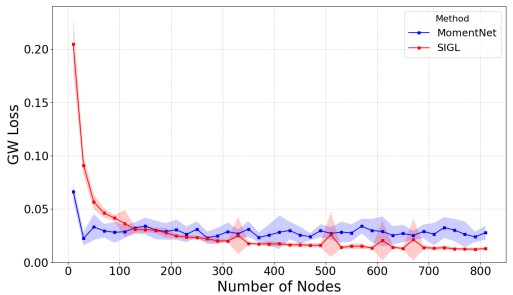
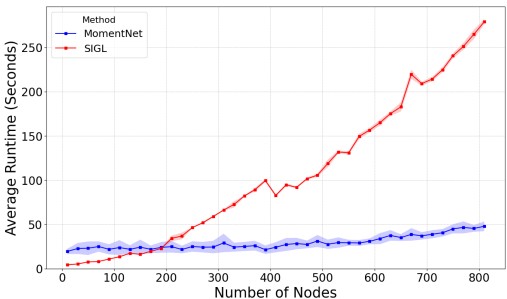

(a) Comparison of performance scalability of Moment-Net with SIGL.

(b) Comparison of runtime scalability of MomentNet with SIGL.

Figure 8: Scalability Comparison of MomentNet and SIGL

## L  MomentMixup Evaluation Details

Our experimental evaluation is conducted on four diverse benchmark datasets widely used in graph classification research. Table 8 provides a detailed overview of these datasets, outlining their specific characteristics and the nature of their respective classification tasks.

Table 8: Description of the benchmark datasets used for evaluation. Each dataset represents a different type of graph structure and classification task.

| Dataset | Description | Classification Task | Citation |
|---------|-------------|---------------------|----------|
| IMDB-B | Movie collaboration graphs; nodes represent actors/actresses, and an edge connects two actors/actresses if they appear in the same movie. | Binary genre classification. | [39] |
| IMDB-M | A multi-class version of IMDB-B, representing movie collaborations with similar graph construction. | Multi-class genre classification. | [39] |
| REDD-B | Social network graphs from Reddit; nodes represent users, and an edge indicates an interaction (e.g., one user commented on another's post). | Binary community (subreddit) classification. | [39] |
| AIDS | Bioinformatics graphs representing molecules; nodes are atoms, and edges are covalent bonds between them. | Binary classification based on anti-HIV activity (active vs. inactive). | [27] |

## M  Social impacts

The methods presented for graphon estimation via moment-matching INRs and data augmentation through MomentMixup, while offering powerful tools for understanding network structures and enhancing graph-based machine learning, are not without potential societal risks if deployed without careful consideration. For instance, in social network analysis, if the empirical moments used for graphon estimation are derived from graphs reflecting existing societal biases (e.g., in representation or connectivity), both the estimated graphons and synthetic graphs generated via MomentMixup could inadvertently perpetuate or even amplify these biases. This could lead to inequitable outcomes when models trained on such data are used for applications like resource allocation, recommendation systems, or public policy modeling. Similarly, in critical domains such as epidemiology or financial systems, inaccuracies in graphon estimation or the generation of unrepresentative augmented data could lead to flawed predictions, potentially resulting in misguided interventions or financial instability. While graphon estimation offers a level of abstraction, careful attention must also be paid to ensure that the process does not inadvertently leak sensitive information from the original graph data, especially when dealing with networks containing personal or confidential details. Therefore,

it is crucial for practitioners to be acutely aware of these potential pitfalls. This includes critically examining input data and chosen moments for biases, rigorously validating the fidelity and representativeness of estimated graphons and generated graphs, and thoughtfully considering the ethical implications of their application, particularly in domains with direct and significant societal impact.

