# OpenReview forum: "A Few Moments Please: Scalable Graphon Learning via Moment Matching"
_NeurIPS.cc/2025/Conference — NeurIPS 2025 poster_

### Official Review · Reviewer_3v3g · 2025-06-26

**Clarity:** 2
**Significance:** 2
**Originality:** 2
**Rating:** 4
**Confidence:** 4

**Summary:**

The paper proposes MomentNet, an implicit-neural-representation (INR) estimator that fits a graphon by matching a fixed set of induced-motif densities (“moments”) from observed graphs.  This approach is claimed to be more scalable and avoids the need for latent variable estimation. They claim a finite-sample bound on the cut distance under an assumption of INR. And they also introduce MomentMixup, a data augmentation technique operating in moment space. Experiments on synthetic graphons and four small graph-classification datasets show modest gains over USVT and IGNR and parity with SIGL.

**Questions:**

1. **Theoretical validity**: Can you provide empirical evidence of when Assumption 1 holds? What are typical motif errors δ achieved in practice, and how do they relate to the theoretical requirements?

2. **Large-scale evaluation**: Can you demonstrate performance on graphs with >5000 nodes to validate scalability claims? Include timing breakdown between Stage 1 (motif counting) and Stage 2 (INR training).

3. **Motif selection**: What is the rationale for using only 4-node motifs when 5-node motifs are available? How does motif dictionary size affect estimation quality and computational cost?

4. **Dense graph performance**: How does the method perform on dense graphs (edge density >0.5) with realistic node counts?

5. **Statistical significance**: Can you provide paired t-tests for MomentMixup improvements and confidence intervals for comparisons?

**Ethical Concerns:**

["NO or VERY MINOR ethics concerns only"]

**Final Justification:**

After the rebuttal and discussion with the author's rebuttal, the author has addressed my concerns about the previous version's mismatch, conducted extra ablation studies on the main issue of complexity. With the author's commitment to including all these fixes and experiment details in the revision, I update my rating from Weak Reject (3) to Weak Accept (4).

**Limitations:**

### **Major Technical Issues**

i) **Assumption 1 Analysis**
   The theoretical guarantee fundamentally depends on the INR’s ability to approximate empirical motif densities within tolerance $\delta$. Refer Weakness 1.

ii) **Complexity Contradiction**
   The paper claims scalability but reveals $\Theta(n^4)$ complexity for dense graphs. Refer Weakness 2. For the dense regime, we have:
   $
   T = O \Bigl(\bigl\lceil P / M\bigr\rceil n_T^4 + N_e \ (L \cdot C_{INR} + |\theta|)\Bigr)
   $
   while SIGL [2] gives
   $
   T = O(P \cdot N_e \cdot n_T^2).
   $
   This makes SIGL asymptotically faster for large dense graphs, contradicting the main scalability claim.

### **Limitations Assessment**

The authors acknowledge some limitations but miss critical issues:

i) **Unaddressed**: Theoretical assumption verification, large-scale evaluation gaps, motif-selection methodology.

ii) **Insufficient**: No discussion of $\Theta(n^4)$ complexity implications or energy consumption for large graphs.

### **Writing Issues**

See Weakness 5: duplicated citations for [27] and [28], wrong reference [22]; unclear symbols.

### **Review Summary**

According to the above weaknesses and limitations, I maintain a Borderline Reject rating as there are several fundamental issues that prevent this work from meeting the NeurIPS standards before being resolved.

### Reference

[1] Xia, X et al. (AISTATS 2023) Implicit Graphon Neural Representation.
[2] Ali Azizpour et al. (AISTATS 2025) Scalable Implicit Graphon Learning.

**Paper Formatting Concerns:**

**Citation Issues**

i) **Duplicate citations**: References [27] and [28] are identical papers with different labels.

ii) **Wrong citation item**: Reference [22] is in wrong format and seems to mix two papers in one item.

**Quality:**

3

**Strengths And Weaknesses:**

### **Strengths**
1. **Clear Idea, Simple Pipeline** : Replacing Gromov-Wasserstein optimisation with direct moment matching is conceptually clean and easy to implement. It reduce optimization overhead present in methods like IGNR [1].
2. **Continuous Representation** – Using an INR yields a resolution-free graphon that can be sampled at an arbitrary size.

### **Weaknesses:**

1. **Vacuous Theoretical Guarantees** I carefully checked the theory and the provided supplementary material. Theorem 1 relies fundamentally on **Assumption 1**, which requires the INR to drive every motif error below $\delta$. However:

	i) No optimization bounds are provided showing when this assumption can be satisfied.

	ii) No sample complexity analysis demonstrates how many samples are needed in practice.

	iii) The condition requires $\Omega(k^2/\delta)$ samples per motif but provides no guidance on achievable $\delta$ values.

2. **Severe Scalability Limitations**  ($O(n^4)$ Dense Complexity) I read and checked the paper and Appendix D. There's a fundamental scalability problem:

	i) **Dense graphs**: Even with full parallelization, Stage 1 costs $\Theta(n^4)$ due to 4-node graphlet counting.

	ii) **Comparison with SIGL**: SIGL achieves $\Theta(n^2)$ complexity, making it asymptotically superior for dense graphs. So the **"Scalable" claim is misleading**: The method only scales well for sparse graphs where degree $d = O(1)$.

3. **Mixed and Questionable Empirical Results**

	i) **Graphon estimation**: MomentNet loses to SIGL on $4/13$ benchmark graphons.

	ii) **Statistical significance**: MomentMixup improvements of $\leq 2$ percentage points fall within $1\sigma$ error bars on most datasets.

	iii) **No ablation studies**: No analysis of motif selection impact or dictionary size effects.

4. **Issuses on Experimental Validation**

	i) **Limited motif analysis**: While ORCA can handle 5-node motifs, authors use only 9 motifs up to 4 nodes without ablation studies on motif dictionary size or selection impact.

	ii)  **Limited scale**: Largest experiment far below the range where the number of nodes becomes prohibitive ($n > 5000$ nodes).

	iii) **Missing dense graph evaluation**: No experiments on realistically large dense networks.

5. **Poor Writing and Citation Issues** During review, I found that,

	i) **Duplicate citations**: References [27] and [28] are identical papers with different labels.

	ii) **Wrong citation item**: Reference [22] is in wrong format and seems to mix two papers in one item.

	ii) **Notation inconsistencies**: Switching between $W$, $W^*$, $\hat{W}_{\theta}$ without clear distinction.

6. **Methodological Concerns**

	i) **Inverse weighting scheme**: The loss function (Eq. 5) uses $w_i = 1/m_i$ weights without theoretical justification, potentially causing instability for small moment values.

	ii) **Limited motif selection guidance**: No principled approach for choosing appropriate moments for different graph types.

	iii) **MomentMixup justification**: While Proposition 1 shows convex combinations of graphons differ from moment combinations, practical implications for data augmentation remain unclear.

### Reference
[1] Xia, X et al. (AISTATS 2023) Implicit Graphon Neural Representation.

[2] Ali Azizpour et al. (AISTATS 2025) Scalable Implicit Graphon Learning.

---

> ### Author Rebuttal · Authors · 2025-07-31
>
> # Reviewer 4
>
> Thanks for reviewing the paper. Answers to your comments and questions follow.
>
> ## Weaknesses and Questions
>
> **1. Vacuous theoretical guarantees and theoretical validity**
> We thank the reviewer for the careful analysis and for raising these important points about our theoretical guarantees. We have included a thorough analysis of the bound in Theorem 1, with empirical evidence showcasing its applicability. Please, refer to the response to reviewer ecYZ for a detailed explanation of this new analysis, that will be included as a new appendix in the camera-ready version of the paper.
>
> **2. Severe scalability limitations**
> Thank you for this detailed review. As stated in the ORCA paper [1], "experiments show that the bound is not tight." In line with this, we conducted empirical experiments and found the actual time complexity of counting motifs up to size four for clique graphs (highly dense) to be $\Theta(n^3)$. Unfortunately, we cannot include our experimental plot due to NeurIPS guidelines preventing to post figures, but we will make sure to include it in the camera-ready version of the paper.
> While the theoretical complexity analysis of SIGL[2] indicates a faster runtime in the dense regime due to its quadratic complexity, our experiments in Figure 2c show that our method's runtime is still faster on larger, dense graphs. This can be attributed to the fact that the empirical time complexity of our method on the dense graphs in our dataset is lower than the theoretical bound and practically comparable to SIGL.
> A key insight from Figure 2 is that our method can subsample a graph to reduce its size while maintaining near-optimal performance, a capability that SIGL lacks. Given the parameters $P$ and $n_T$ in our time complexity, we can decrease $n_T$ in highly dense regimes to drastically reduce the runtime while ensuring a sufficient $P$ to preserve performance. Although not explicitly shown in Figure 2c, this can be easily deduced. Therefore, our claim of scalability remains valid even for highly dense graphs. We will add a comprehensive experiment on this trade-off to the appendix of the camera-ready version.
>
> **Large-scale evaluation**
> The same argument mentioned in the previous part holds for experiments on graphs with more than 5000 nodes. Since our model does not require the full graph as input to the INR to perform well, we can leverage subsampling and decrease the $n_T$ parameter. This strategy allows us to achieve a significantly better runtime and maintain strong performance compared to SIGL. We will demonstrate this empirically in the comprehensive experiment mentioned previously. For the sake of the rebuttal, we present a focused experiment to specifically validate our claim on large graphs where scalability is a key challenge. The experiment is detailed in our response to Reviewer ecYZ's concern regarding the computational cost compared with SIGL.
> A extended version of this experiment will be included in the camera-ready version of the paper, if it gets accepted.
>
> **Dense graph performance**
> In response to the reviewer's question about graphon estimation with dense graphs, we want to emphasize that our experiments already cover dense graphons. Specifically, Graphons 3, 4, 5, 6, 9, and 11 in Table 2 of the appendix are all dense graphs with an edge density greater than 0.5. The corresponding results are shown in Figure 2a. Additionally, the graphon used for the scalability analysis in Figures 2b and 2c is also dense. If the reviewer is interested in a particular graph structure that we have not addressed, we would appreciate the clarification.
>
> **3. Mixed and questionable empirical results**
> **3.i) MomentNet loses to SIGL on 3/14 benchmarks**
> We acknowledge the reviewer's point regarding the 3 (out of 14) benchmark graphons where SIGL shows a slight performance advantage. We wish to place this result in the proper context:
> * **Competitive Performance:** As shown in Figure 2a, the performance difference on these few benchmarks is minimal. Conversely, MomentNet demonstrates superior or equivalent performance on the vast majority (11/14) of the benchmarks, showcasing its strong generalizability.
> * **A Favorable Trade-off for Scalability:** When analyzing large graphs sampled from any given graphon class, MomentNet has the ability to converge on a high-accuracy solution significantly faster than competing methods.
> In summary, while SIGL may attain a marginal victory on a small subset of the graphons, MomentNet provides a more compelling overall package: state-of-the-art accuracy in most cases and superior scalability in all cases, rendering it a more versatile and practical method.
>
> **3.ii) Statistical significance**
> Although our focus was not on establishing a new state-of-the-art in graph classification, we aimed to provide a novel, scalable, and flexible alternative for the challenging task of graph data augmentation using mixup. As demonstrated, our results serve as a proof-of-concept that validates this as a viable approach to the graph data augmentation problem. We will make sure to relax the statements in this regard for the camera-ready version of the paper. It is also worth mentioning that the marginal gain in both g-mixup and SIGL is similarly small compared to previous SOTA methods. This is primarily because these methods rely solely on the graph structure for creating new samples and for classification; incorporating node features is not yet possible. As we state in the future work section of the conclusion, we aim to investigate how to incorporate features into this graph generation process, which will enable us to use a wider range of datasets. We plan to conduct statistical tests to validate the results and will add them to the paper in the next revision.
>
> **3.iii) Ablation studies and motif selection**
> We thank the reviewer for the careful analysis and for raising these important points about our theoretical guarantees. We have included a thorough ablation test. Please, refer to the response to reviewer ogzT for a detailed explanation of this new analysis, that will be included as a new appendix in the camera-ready version of the paper.
> Regarding the computational complexity of higher-order motifs, our approach remains efficient. Even for graphons that are highly dependent on these more complex structures, our subsampling strategy allows for strong performance with a significantly smaller runtime compared to other baselines. We will add this experiment and the statement about complexity in the large-graph regime to the camera-ready version of the paper.
>
> **4. Issues on experimental validation**
> * **Limited motif analysis**: The ablation study is provided in the previous subsection. We also want to emphasize that our method is agnostic to methods for counting graphlets.
> * **Limited scale**: We addressed this potential weakness in Section 2 of our response.
> * **Missing dense graph evaluation**: We addressed this potential weakness in Section 2 of our response.
>
> **5. Poor writing and citation issues**
> We apologize and thank you for highlighting these issues. They will be corrected in the camera-ready version, which will be carefully proofread.
>
> **6. Methodological concerns**
> * **Inverse weighting scheme**: We appreciate your concern regarding the potential instability of our approach. As demonstrated by our results for sparse graphons in Figure 2a, the training process is indeed stable. To further mitigate instability, particularly in cases of very low graph densities, we set the corresponding weights to 1. This weighting scheme was selected after an empirical evaluation against several alternatives (no weighting or exponential weighting schemes), where it demonstrated superior performance. We will clarify the details of this strategy in the appendix of camera-ready version of the paper.
> * **Limited motif selection guidance**: We addressed this potential weakness in Section 3 of our response.
> * **MomentMixup justification**: The practical implication of our approach is to introduce a data augmentation technique for graphs analogous to mixup[3] in other domains like computer vision. Just as mixup creates new training samples by interpolating between existing data points and their labels to improve generalization, our method of creating convex combinations of graphons generates novel, synthetic graphs. This encourages the model to learn a smoother, more robust decision boundary. Our rationale for using moment mixup over g-mixup[4] is to ensure that augmented samples remain structurally similar to a specific class. We define this structural proximity through subgraph densities, which are precisely captured by graph moments. As established in Proposition 1, performing mixup directly on graphons does not guarantee the preservation of these crucial moment-based structures.
>
> Thank you again for your valuable feedback and suggestions, which we believe have contributed to clarifying and strengthening our work.
>
> ## References:
> [1] T. Hocevar, et al. (Bioinformatics 2014) A combinatorial approach to graphlet counting
>
> [2] A. Azizpour et al. (AISTATS 2025) Scalable Implicit Graphon Learning
>
> [3] H. Zhang, et al. (ICLR 2018) mixup: Beyond Empirical Risk Minimization
>
> [4] X. Han, et al. (PMLR 2022) G-Mixup: Graph Data Augmentation for Graph Classification

---

> > ### Comment · Reviewer_3v3g · 2025-08-05
> >
> > Thank you for your detailed responses and efforts. I have checked your new reply and paper. However,  there exist some new issues and mismatch of your paper and your rebuttal,
> >
> > **Technical Contradictions**: Your response contains a direct contradiction regarding complexity analysis. **In your Appendix D clearly states dense graphs require** $\Theta(n^4)$ **complexity, yet your rebuttal cite other paper and claims empirical results show** $\Theta(n^3)$. This inconsistency indicates confusion about your own method's computational characteristics. Furthermore, your subsampling argument fundamentally misunderstands the scalability problem—reducing graph size from 2000 to 50 nodes is not solving large-scale processing but rather avoiding it entirely. "we conducted empirical experiments and found the actual time," you using the $\Theta$ asymptotic complexity notation, which is not sufficient to valided from empirical experiments, if you using $\Theta$, you should give clear analysis for the order of the asymptotic constant.
> >
> > I maintain my score and hope you can give futher clear jusfication.

---

> > > ### Author Response · Authors · 2025-08-05
> > > **Addressing additional concerns from Reviewer 3v3g**
> > >
> > > We first want to thank you for your engagement in the discussion, which surely helps us improve the quality of the paper.
> > >
> > > **ORCA**
> > >
> > > We appreciate the reviewer’s careful reading and insightful comments on the complexity analysis. As we are using the ORCA algorithm [1], our statement about the complexity not being a tight bound was a direct reference to the findings of the original authors, which we also observed empirically.
> > >
> > > Regarding the notation, we thank the reviewer for highlighting this. To clarify, the appendix states the theoretical worst-case time complexity of ORCA in Big O notation, per the source paper [1]:
> > > $$T_{\text{ORCA}}(G_p)=O\bigl(e_p d_p + n_p d_p^{3}\bigr)$$
> > > For a dense graph, this theoretical bound is $O(n_p^4)$. The reviewer's concern about our use of $\Theta$ in the rebuttal is well-founded, as it was an imprecise description of the empirical results. We address this in detail in the final section of this comment.
> > >
> > > **Scalability Experiment**
> > >
> > > We appreciate the opportunity to clarify our methodology for the scalability experiment. We acknowledge that the subsampling technique is employed as a practical strategy to manage runtime. However, two important aspects to consider are the following. First, we do not replace the full graph with a smaller one but use it to generate multiple smaller graphs, each of them formed by sampling nodes from the larger input graph and keeping the induced edges. Second, since we assume that the input graph is generated from a specific graphon, the subsampled graphs can be interpreted as realizations drawn from the same graphon, providing additional support to our method.
> > >
> > > As our scalability figures (Figures 2b, 8a) demonstrate, our method achieves performance comparable to competing approaches once the subsampled graph is sufficiently large. We wish to highlight that our method is designed to operate effectively in this small-graph regime. This approach is advantageous as it can better approximate the underlying graphon compared to methods like SIGL [2], while avoiding the high computational cost of analyzing entire large graphs, a common limitation in this field. To be clear, we do not suggest that counting motifs in a large graph is not cumbersome; we emphasize that our proposed method is designed specifically to reduce this burden.
> > >
> > > **4-node Graphlet Counting Bound**
> > >
> > >
> > > Thank you for your insightful feedback regarding the presentation of our empirical results. We agree that using theoretical notation such as $\Theta$ for describing experimental findings is imprecise. We appreciate the opportunity to provide the full details of the experiment we referenced, which were omitted from the previous response due to space constraints.
> > >
> > > **Details of the Empirical Runtime Experiment**
> > >
> > > The experiment was designed to precisely characterize the practical runtime of ORCA[1] on dense graphs.
> > >
> > > * **Methodology:** We measured the wall-clock execution time for counting 4-node graphlets on a sequence of dense graphs of varying sizes. The number of nodes, $n$, was varied from 30 to 400. To model the relationship between runtime $T(n)$ and node count $n$, we performed a linear regression on the log-transformed data points.
> > >
> > > * **Empirical Performance Model:** The analysis confirmed a strong polynomial relationship, yielding the following data-driven model:
> > >     $$T(n) \approx c \cdot n^k$$
> > >     Our results show an exponent of **$k \approx 3$**, confirming a near-cubic growth rate in practice for these graphs. Crucially, as requested, our analysis determined the constant factor of ORCA to be **$c \approx 2.97 \times 10^{-8}$**.
> > >
> > > * **Goodness of Fit:** The model provides an excellent fit to our measured data, evidenced by a high coefficient of determination of **$R^2 \approx 0.97$**.
> > >
> > > This detailed model provides the rigorous, quantitative evidence for our claim about the method's practical performance. It demonstrates that the algorithm is highly efficient on dense graphs due to the extremely small leading constant.
> > > In conclusion, and to address the concern about scalability, our method is faster when compared to another scalable approach, SIGL. This practical speed is due to the small constant in our method's empirical time complexity. We did not determine the tight theoretical upper bound for ORCA, as it was outside the scope of our main goal. Furthermore, based on Theorem 1 and our problem definition, both the number of nodes and the number of graphs are tunable parameters for estimating the graphon. As shown in the performance plot in Figure 2b, we claim that given a large graph, we can subsample smaller graphs and preserve good performance, a capability that SIGL lacks.
> > >
> > > We will update the paper and include these specific details in the appendix of the camera-ready version to ensure full clarity.
> > >
> > > [1] T. Hocevar, et al. (Bioinformatics 2014) A combinatorial approach to graphlet counting
> > >
> > > [2] A. Azizpour et al. (AISTATS 2025) Scalable Implicit Graphon Learning

---

> > > > ### Comment · Reviewer_3v3g · 2025-08-05
> > > >
> > > > Thanks for your details clarification and efforts. By now, all my concerns have been resolved, and I hope you can include these extra details in your future revision. I will increase my rating score in my final justification~

---

> > > > > ### Author Response · Authors · 2025-08-05
> > > > > **Thank you for your thorough review**
> > > > >
> > > > > Thank you for your helpful feedback and suggestions. We are happy that our clarifications addressed all of your concerns. We will add these details to the revision and greatly appreciate your support and increased score.

---

### Official Review · Reviewer_ecYZ · 2025-06-29

**Clarity:** 3
**Significance:** 3
**Originality:** 3
**Rating:** 4
**Confidence:** 4

**Summary:**

This paper proposes MomentNet, a scalable graphon estimator that employs moment matching with implicit neural representations (INRs). The method estimates graphons by matching empirical subgraph counts ("moments") derived via counting from observed graphs. The authors demonstrate theoretically that when the neural network can accurately approximate the motifs, the overall graphon estimation error remains bounded. Through moment matching, they circumvent computationally expensive Gromov-Wasserstein loss estimations.

**Questions:**

- Can you provide concrete examples where the computational advantage over SIGL becomes critical?
- How sensitive is the method to Monte Carlo sampling variance during training?
- Can you compute the actual bounds from Theorem 1 for your experimental settings?

**Ethical Concerns:**

["NO or VERY MINOR ethics concerns only"]

**Final Justification:**

While I still have some doubts about the practical relevance of the polynomial runtime and the experimental gains, I acknowledge that choosing all motifs with up to 4 nodes might be a good general choice of considered motifs and I consider this as sufficiently resolved. I particularly like the theory of the paper and that it proposes a novel approach instead of something totally incremental.

**Limitations:**

While not covered in an explicit "Limitations" section, this is honestly discussed in the Conclusion.

**Paper Formatting Concerns:**

Appendix is in a separate file, not in the main PDF.

**Quality:**

3

**Strengths And Weaknesses:**

## Strengths:

**Overall:**

The central insight connecting motif approximation to graphon estimation quality (Theorem 1) is elegant and well-motivated. The idea of bypassing expensive Gromov-Wasserstein optimization through direct moment matching represents a meaningful algorithmic contribution. The paper is generally well-written.

**Clarity** while good in general could be improved at two points:
- ORCA and graphlets are fundamental to the approach but barely explained.
- Strange wording: l14-15 “the estimated graphon achieves a provable upper bound in cut distance from the ground truth”

**Reproducibility:** Code is provided and results seems reproducible

## Weaknesses:

**Critical moments-selection limitation not addressed:**
The authors acknowledge that relying on pre-selected moments is a major limitation but defer this to future work. While I appreciate this honest discussion, for a venue like NeurIPS with its strong practical focus, this seems like a fundamental issue that should be addressed in the current work. Without adaptive moment selection, the method's applicability remains quite limited, and it raises concerns about potential cherry-picking of motifs. This limitation significantly reduces the paper's practical significance, as the effectiveness of the approach becomes heavily dependent on domain expertise for motif selection. The contribution would be substantially more impactful if an automatic moment selection procedure were introduced and demonstrated to work well across different graph types.

**Limited practical impact of computational gains:**
While the polynomial runtime is nice theoretically, the experiments show SIGL running in ~250 seconds, which isn't prohibitive for most applications. The authors should demonstrate scenarios where existing methods become truly infeasible to strengthen their computational efficiency claims.

**Missing technical validation:**
No discussion of Monte Carlo gradient estimation stability, which could be problematic for training.
The theoretical bounds in Theorem 1 are never computed for the experimental settings. Showing whether these bounds are tight would significantly strengthen the claims.

## Summary:
While the core idea has merit and Theorem 1 provides compelling theoretical motivation, the paper needs to address the fundamental moment selection problem and demonstrate clearer practical advantages before meeting the bar for NeurIPS. The theoretical contributions, though heavily building upon existing work, are sound and well-executed. The experimental gains are modest but show promise for the approach's potential.

---

> ### Author Rebuttal · Authors · 2025-07-31
>
> Thanks for reviewing our paper and the feedback you provided. We address your comments and questions in the following subsections:
>
> **1. Clarity issues.**
>
> ORCA and graphlets: As our proposed method is independent of the specific algorithm for graphlet and motif enumeration, we have adopted the ORCA toolkit for our implementation. A thorough explanation of the ORCA methodology, along with a detailed explanation of graphlets, is available in the original publication [1], to which we referred the interested readers. We will include a more detailed summary of these concepts in the final manuscript, if accepted.
>
>  Strange wording; "the estimated graphon achieves a provable upper bound in cut distance from the ground truth". We thank the reviewer for spotting this. We modified it so that the sentence in the camera-ready version will read as: "the cut distance between the estimated graphon and the ground truth is provably upper-bounded."
>
> **2. Ablation test for Moment selection.**
> We thank the reviewer for the careful analysis and for raising these important points about our theoretical guarantees. We have included a thorough ablation test. Please, refer to the response to reviewer ogzT for a detailed explanation of this new analysis. This analysis will be included as a new appendix in the camera-ready version of the paper.
>
>
> **3. Computational cost concern compared with SIGL.**
>
> The scalability evaluation in Figure 2 (Section 5.1.2) demonstrates the conditions under which our method achieves a decisive computational advantage over SIGL. A key insight from both the figure and a similar experiment in the appendix is that SIGL’s performance deteriorates on smaller graphs. This limitation benefits our approach: by subsampling a very large graph into a smaller, more manageable subgraph, we still outperform SIGL. In contrast, SIGL would require well over 250 s to obtain comparable results on the same subgraph, making such sampling ineffective for that method. Although this runtime is not explicitly plotted, it can be inferred from the reported data (this will be clarified in the camera-ready paper).
>
> To further validate this observation, we constructed a dataset of ten graphs, each containing 2,000 nodes. Because both MomentNet and SIGL incur high computational costs when processing the full graphs, we extracted ten 50-node subgraphs from each 2,000-node graph and used these as inputs. We adopted the same dense-graph graphon class employed in the scalability analysis of MomentNet (Section 5.1.2). The results, summarized in the table below, confirm that our method outperforms SIGL in both accuracy and runtime in the large-graph regime. Crucially, while SIGL’s runtime increases sharply with graph size, our method maintains comparable performance even on the smaller subgraphs. A more comprehensive version of this experiment will be included in the camera-ready paper.
>
> | Method | Training Runtime (s) | GW Loss | Motif Counting Runtime (s) |
> | :--- | :---: | :---: | :---: |
> | **MomentNet** | $54.83 \pm 18.69$ | $0.0548 \pm 0.0016$ | $1.59 \pm 0.24$ |
> | **SIGL** | $207.89 \pm 18.4$ | $0.1085 \pm 0.0156$ | - |
>
> **4. Can you compute the actual bounds from Theorem 1 for your experimental settings? The theoretical bounds in Theorem 1 are never computed for the experimental settings. Showing whether these bounds are tight would significantly strengthen the claims.**
>
>
> We thank the reviewer for this very valuable suggestion. The primary purpose of our theoretical section is to provide a formal justification for our method's design and to formalize the intuition that its performance improves with access to more or larger graph samples. We argue that our guarantees, while relying on a standard assumption, are valid, as the assumption is empirically validated and the theoretical conditions are met in practical scenarios. We will cover next the different aspects and assumptions of the proposed theoretical characterization.
>
> *a. On Satisfying Assumption 1 (Achievable Motif Error)*
>
> The reviewer is correct that we do not provide a formal optimization guarantee. However, Assumption 1 is standard in literature involving neural network estimators and is supported by two pillars: the Universal Approximation Theorem (guaranteeing a sufficiently expressive network *exists*) and the widespread empirical success of modern optimizers in finding high-quality solutions.
>
> While a formal proof is challenging, we demonstrate empirically that Assumption 1 is readily satisfied. Our loss function, $L(\theta)$, is a direct proxy for the motif approximation error required by the assumption. The table below shows the final average loss for a benchmark graphon, using motifs up to size $k=4$ (9 total motifs).
>
> |#Motifs|avg_loss|std_loss|
> |---|---|---|
> |1|0.0|1e-06|
> |2|2e-06|1e-06|
> |3|1.4e-05|1.5e-05|
> |4|5e-06|3e-06|
> |5|1e-05|1e-05|
> |6|2e-05|1.4e-05|
> |7|3e-05|2.5e-05|
> |8|5.4e-05|3.7e-05|
> |9|5.8e-05|6.7e-05|
>
> The optimization consistently drives the motif error to extremely small values (on the order of 1e-5), showing that achieving a small $\delta_M$ is an empirical reality for our method, not just a theoretical hope.
>
> *b. On Sample Complexity and necessary probabilistic Bounds*
>
> Given that a small motif error $\delta_M$ is achievable, we now show that the overall probabilistic bound from Equation (6) is non-vacuous for realistic dataset sizes. We analyze this bound for $k=4$ and a conservatively large motif error of $\delta_M = 0.07$ (which is orders of magnitude larger than our empirical results). The table below shows the failure probability (lower bound of $\zeta$ in equation 6) for various numbers of graphs $P$ and nodes $n$.
>
> |$\zeta$|$P=400$|$P=600$|$P=800$|$P=1000$|$P=1200$|$P=1400$|$P=1600$|$P=1800$|$P=2000$|
> |---|---|---|---|---|---|---|---|---|---|
> |$n=200$|11.63|11.45|11.27|11.10|10.93|10.76|10.59|10.43|10.26|
> |$n=250$|9.93|9.04|8.22|7.48|6.81|6.19|5.63|5.13|4.66|
> |$n=300$|7.87|6.37|5.16|4.18|3.39|2.74|2.22|1.80|1.46|
> |$n=350$|5.97|4.22|2.97|2.10|1.48|1.04|0.74|0.52|0.37|
> |$n=400$|4.42|2.68|1.62|0.99|0.60|0.36|0.22|0.13|0.08|
> |$n=450$|3.21|1.66|0.86|0.44|0.23|0.12|0.06|0.03|0.016|
>
> Although values > 1 are uninformative as probabilities, the table clearly shows the bound's exponential decay. Crucially, the guarantee becomes meaningful for realistic data regimes. For example, the REDDIT-B dataset has $P=2000$ graphs with an average $n \approx 497$. Our analysis shows that in a comparable setting ($n=450$, $P=2000$), the failure probability $\zeta$ is a practically useful **0.016**. This confirms the bound is not vacuous.
>
> *Conclusion*
>
> In summary, our theoretical analysis provides a sound foundation for our method by formally connecting the quantity of input data ($P$ and $n$) to the final estimation error. We show that:
> 1.  The core assumption of learning motifs accurately is empirically validated.
> 2.  The theoretical conditions for our guarantee to hold are met within realistic data regimes.
>
> Therefore, the theory serves its purpose of providing a formal rationale for our moment-matching approach. We will add an appendix clarifying this perspective and summarizing these empirical findings to strengthen the paper.
>
> **5. Monte Carlo sampling variance.**
> Based on the relatively high quality of the results achieved by our method, we hypothesize that stable convergence is contingent upon a low Monte Carlo variance. To investigate this hypothesis directly, we designed an experiment to measure the gradient variance as a function of the number of MC samples. For this test, the parameters of the INR model were held constant while we estimated the gradient's standard deviation over 1,000 runs for varying numbers of samples, $L$. The results, summarized in the following table, show a clear inverse relationship: the standard deviation consistently decays as $L$ increases. This finding supports our hypothesis and underscores the importance of using a sufficient number of samples. We will add this experiment to the appendix of the camera-ready version of paper.
>
> | $L$    | mean_gradient | std_gradient |
> | :----- | :------------ | :----------- |
> | 100    | 0.013005      | 0.004484     |
> | 500    | 0.013107      | 0.002031     |
> | 1000   | 0.013030      | 0.001387     |
> | 5000   | 0.013019      | 0.000647     |
> | 10000  | 0.013052      | 0.000439     |
> | 20000  | 0.013009      | 0.000325     |
> | 50000  | 0.013013      | 0.000193     |
> | 100000 | 0.013019      | 0.000134     |
>
>
> We conclude by thanking the reviewer for feedback and suggestions that, in our view, have contributed to clarify our contributions and strengthened the paper.
>
> ## References:
> [1] T. Hocevar, et al. (Bioinformatics 2014) A combinatorial approach to graphlet counting

---

> > ### Comment · Reviewer_ecYZ · 2025-08-01
> >
> > Dear authors, thanks for taking time and your genuine effort for discussing and resolving the issues of my reviews. The ablation study as well as the Monte Carlo variance analysis and the theoretical bounds computation give useful insights.
> >
> > While I still have some doubts about the practical relevance of the polynomial runtime and the experimental gains, I acknowledge that choosing all motifs with up to 4 nodes might be a good general choice of considered motifs. Also, I think the theory of the paper is really cool and future research might build upon this novel approach.
> >
> > I will increase my overall score from borderline reject to borderline accept and the quality and significance score from 2 to 3.

---

> > > ### Author Response · Authors · 2025-08-05
> > > **Thank you for your thorough review**
> > >
> > > We sincerely thank you for your time. We are very happy that our additional experiments, including the ablation study and Monte Carlo analysis, helped resolve your earlier concerns.

---

### Official Review · Reviewer_K8VW · 2025-07-03

**Clarity:** 3
**Significance:** 3
**Originality:** 2
**Rating:** 4
**Confidence:** 3

**Summary:**

This paper proposes MomentNet, a novel and theoretically principled approach for graphon estimation using implicit neural representations (INRs) trained via moment matching. It avoids costly latent variable inference and combinatorial Gromov-Wasserstein (GW) alignment by learning from empirical motif statistics extracted from observed graphs. The authors further introduce MomentMixup, a graph data augmentation strategy that interpolates motif vectors and generates synthetic graphs via the learned graphon.
The paper is well-structured and contains meaningful theoretical insights. It also reports performance improvements in both graphon estimation (on synthetic benchmarks) and graph classification (on small real datasets). However, I believe the paper in its current form does not provide sufficient experimental validation or analysis to support its claims fully.

**Questions:**

Suggestions for Improvement
1. Include ablation experiments that isolate the effect of MomentNet (without mixup) vs. MomentMixup (with simple generators).
2. Perform motif selection analysis, including importance ranking, sensitivity motif size/type, and robustness to noise.
3. Test MomentNet on more complex graphons (e.g., high-frequency, oscillating, or hierarchical structures).
4. Try estimating graphons from real-world graphs and visualizing or interpreting the learned functions.
5. Add stronger augmentation baselines such as DropEdge, GraphTransplant, or SubgraphMix, to better situate MomentMixup.

**Ethical Concerns:**

["NO or VERY MINOR ethics concerns only"]

**Final Justification:**

This paper proposes MomentNet, a theoretically grounded approach for graphon estimation using implicit neural representations (INRs) trained via moment matching. It eliminates the need for costly latent variable inference and Gromov-Wasserstein alignment by learning from motif statistics in observed graphs. The authors also introduce MomentMixup, a graph data augmentation method that interpolates motif vectors to generate synthetic graphs via the learned graphon.

The authors have made valuable improvements, especially with the ablation studies and motif selection analysis, which address key concerns raised by the reviewers. Although I still have some doubts, such as the insufficient baseline comparison, I believe the interesting theoretical contributions make the paper potentially acceptable.

**Limitations:**

yes

**Quality:**

2

**Strengths And Weaknesses:**

**Strengths**
1. Theoretically grounded: The paper offers a rigorous upper bound on the cut distance between estimated and true graphons using a moment-based INR approximation, which is both novel and elegant.
2. Conceptual simplicity: The use of empirical motifs to bypass latent node estimation is clever, computationally efficient, and scalable.
3. Clear writing and organization: The overall exposition is solid and follows logical structure from motivation to theory to experiments.

**Weaknesses**
1. Weak Evaluation of MomentMixup
While MomentMixup is a key contribution of the paper, the empirical support is limited: Only one baseline is considered (G-Mixup); Only four relatively small datasets are used (IMDB-B, IMDB-M, REDDIT-B, AIDS); The performance improvements presented are marginal in most settings.
2. Fixed and Unanalyzed Motif Design
The method relies on a fixed set of motifs (up to 5 nodes) computed using the ORCA library, but provides no justification, selection criteria, or ablation analysis. The performance may heavily depend on motif relevance, redundancy, or sparsity. This weakens the generality and robustness of the proposed method, especially when applied to diverse graph types.
3. Lack of Graphon Complexity in Experiments
All graphon estimation experiments are conducted on relatively smooth, synthetic graphons. The INR used is a shallow 1-layer MLP. There is no investigation into whether MomentNet can recover more complex or high-frequency graphons, or how its performance scales with harder tasks. This limits the scope and applicability of the method.

---

> ### Author Rebuttal · Authors · 2025-07-31
>
> Thank you for your comments and feedback. We address these in the following subsections:
>
> **1. Graphon complexity and INR architecture**
> We evaluated our method using stochastic block models and a bipartite graphon (items 12 and 13 in the appendix, Table 2) as hierarchical graph representations, as well as a cosine graphon as an oscillating instance for the scalability evaluation. The results of these experiments, presented in Figure 2, show that our method's performance was superior to SOTA approaches.
>
> We agree that while a simple architecture can perform well on benchmark datasets, a more complex network is generally required to represent a highly complex function. This is why we treated the network as a hyperparameter (which we will make sure to clarify in the manuscript, as it is not there now), and select the best performing architecture for each graphon. This reflects the known limitations of modeling complex functions with small or shallow neural networks. Note that this is also true for other approaches that use INRs.
>
> During the rebuttal, we did not have time to run experiments tailored to your comment, but we will revise the paper to incorporate the points you raised and a summary of this response.
>
> **2. Motif selection analysis**
> We thank the reviewer for the careful analysis and for raising these important points about our theoretical guarantees. We have included a thorough ablation test. Please, refer to the response to reviewer ogzT for a detailed explanation of this new analysis, which will be included as a new appendix in the camera-ready version of the paper.
>
>
> Regarding robustness to noise:
> The generation of a graph from a graphon is an inherently random process (as detailed in the Graphons subsection of Section 2). Consequently, the moment vector computed from a single sampled graph is a random (noisy) observation of the true moment vector. By generating graphons independently and averaging over their associated vectors, we obtain an unbiased estimator of the actual moments.
> Therefore, the experiments we conducted inherently used these random observations of the true moment vector. Introducing other kinds of noise to the graphon or the graph sampling process is of interest but, unlike this statistical noise, would risk making the estimator biased.
>
> **3. Other data-augmentation baselines and evaluation of MomentMixup.**
>
> *Baselines:* We compared our method against G-Mixup [1], a SOTA approach. In their paper, the authors show that G-Mixup is superior to other methods like DropEdge and Subgraph Mixup (see Table 2 in [1]). Since G-Mixup represents a stronger and more relevant baseline, we focused our comparison on it. Including the scores for these weaker baselines would not alter the conclusions drawn from our results.
>
> *Evaluation of MomentMixup:* The limitations regarding the datasets are primarily because our method relies on structural proximity. Incorporating graph features into the graphon estimation process is left as future work, which will enable us to conduct experiments on a broader range of datasets. Even though some of the selected datasets are relatively small, our scalability analysis (Figure 2b, c) shows that our method outperforms others in graphon estimation. The results in Table 1 verify this claim. More specifically, for the AIDS dataset, which lacks large graphs, other approaches degraded in performance due to inaccurate graphon estimation, whereas our method excelled. The details of the datasets are provided in the following table. It is also worth mentioning that other methods are unable to estimate the graphon based on the moment vector and thus are not able to satisfy Proposition 1.
> | Dataset         | Total Graphs | Min Nodes | Max Nodes |
> |-----------------|--------------|-----------|-----------|
> | IMDB-BINARY     | 1000         | 12        | 136       |
> | IMDB-MULTI      | 1500         | 7         | 89        |
> | REDDIT-BINARY   | 2000         | 6         | 3782      |
> | AIDS            | 2000         | 2         | 95        |
>
> Furthermore, the marginal gains observed in data augmentation can be attributed to our focus on structural proximity, deliberately avoiding the incorporation of other factors like node features in this process. We plan to explore the integration of such features in our future work. We want to emphasize that our primary contribution is the estimation of scalable graphons using moments, which enables applications such as data augmentation and statistical network analysis, like the centrality measures mentioned in the appendix. A summary of this response will be incorporated in the revised paper.
>
> **4. Visualizing real-world graph distribution.**
> As mentioned in the conclusion (line 361), we hypothesize that in real-world datasets, graphs within each class do not belong to a single specific distribution but rather to a mixture of graphons. In this paper, we focused on homogeneous graphs for the graphon estimation task. To address this comment, we have performed a preliminary experiment on moment-based clustering, to be able to create graphs from a class coming from a mixture of graphons. We couldn't report these preliminary results here due to the impossibility to include figures. If the paper gets accepted, we will expand the experiments and include them in the camera-ready version of the paper. This approach is effective because graphs generated from the same underlying graphon have very similar moment vectors. Consequently, when represented as points in the moment feature space, these graphs naturally form distinct clusters.
>
>
> Thank you again for your feedback and suggestions, which we believe have clarified and strengthened our work.
>
> ## References:
> [1] X. Han, et al. (PMLR 2022) G-Mixup: Graph Data Augmentation for Graph Classification

---

> > ### Comment · Reviewer_K8VW · 2025-08-03
> >
> > Thanks for your detailed and thoughtful response. You have made valuable improvements, especially with the ablation studies and motif selection analysis, which address key concerns raised by the reviewers.
> >
> > I will increase my overall score from borderline reject to borderline accept.

---

> > > ### Author Response · Authors · 2025-08-05
> > > **Thank you for your thorough review**
> > >
> > > Thank you for your positive feedback and for re-evaluating our work. We are glad that the improvements addressed your concerns. We sincerely appreciate your support and your decision to increase our score.

---

> ### Comment · Reviewer_K8VW · 2025-08-06
>
> You're welcome. Additionally, note that for the paper you referenced [1], a report [2] seemingly indicates that the G-Mixup may not be as advantageous as claimed by the authors. A 2025 survey [3] also summarizes newer methods that have emerged in the field.
>
> [1] Han, Xiaotian, et al. "G-mixup: Graph data augmentation for graph classification." International conference on machine learning. PMLR, 2022.
>
> [2] Omeragic, Ermin, and Vuk Đuranović. "[Re] G-Mixup: Graph Data Augmentation for Graph Classification." NeurIPS 2023, https://neurips.cc/virtual/2023/poster/74183.
>
> [3] Zhou, Jiajun, et al. "Data augmentation on graphs: A technical survey." ACM Computing Surveys 57.11 (2025): 1-34.

---

> > ### Author Response · Authors · 2025-08-06
> > **Thank You for the Helpful References**
> >
> > Thank you for sharing these helpful articles! We appreciate you pointing that out about G-Mixup, and we will be sure to add a discussion about its limitations, using the reference you provided.
> > The survey paper is also very useful. We'll look through it for this paper and keep it in mind for our future work. We really appreciate your help!

---

### Official Review · Reviewer_ogzT · 2025-07-03

**Clarity:** 4
**Significance:** 3
**Originality:** 3
**Rating:** 5
**Confidence:** 3

**Summary:**

This paper proposes MomentNet, a novel, scalable graphon estimation method using moment matching and implicit neural representations (INRs). Instead of inferring latent positions or minimizing costly distances like Gromov-Wasserstein (GW), it uses motif count statistics (graphon moments) as supervisory signal. The key idea is to optimize an INR to match empirical subgraph densities from observed graphs. The authors further propose MomentMixup, a mixup-based data augmentation strategy in moment space, which enables structural interpolation of graphs via synthetic moment vectors decoded through MomentNet.

The paper offers a new path in graphon estimation by combining solid theoretical foundations (e.g., cut-distance guarantees via inverse counting lemma) with practical scalability. It demonstrates strong empirical results on synthetic graphons and several real-world graph classification datasets.

**Questions:**

1. *On Moment Selection*: Can you elaborate on how robust your method is to the choice or number of motifs? Would adaptive motif selection (e.g., using entropy or redundancy) improve performance?

2. *On MomentMixup Semantics*: When combining moment vectors from two classes, how is the label assigned to the generated graph? Is it a soft label, or do you assign based on dominant α weight?

3. *On INR Generalization*: Have you evaluated whether the trained MomentNet generalizes well to moment vectors outside the training distribution (e.g., interpolations or extrapolations)? How stable is the mapping from moments to graphons?

4. *On Heterogeneous Graph Classes*: How would your method adapt to datasets with significant intra-class structural variability? Could a mixture-of-graphons model address this?

5. *On Motif Sampling Efficiency*: The runtime and sampling complexity of computing motif densities (especially with ORCA) might still be high. Could you consider approximating them via graphlet sampling or neural surrogates?

**Ethical Concerns:**

["NO or VERY MINOR ethics concerns only"]

**Final Justification:**

I was initially concerned about the practicality of the proposed approach. However, the authors have addressed these concerns convincingly:

*Motif Count*: The use of five motifs in the experiments is justified by an ablation study, which demonstrates that incorporating more than six motifs does not yield further performance improvements.

*Generalization*: The model’s ability to generalize has been validated by training on 9 motifs and extrapolating to 20 unseen motifs, with strong performance—relative errors are consistently low (typically <4%).

*Heterogeneous Graph Classes*: For these cases, the authors propose partitioning graphs into structurally homogeneous clusters using moment vectors. Their claim—that graphs from the same generative mode have similar moments and are easily separable—provides a sound theoretical basis. While experimental validation is pending, the authors have committed to conducting these tests.

Given these clarifications and the strength of the ablation and extrapolation results, I believe the paper’s contributions are more robust than I initially assessed and I am inclined to raise my score.

**Limitations:**

The authors acknowledge key limitations in Section 6, including:

  - Sensitivity to the choice of moments.

  - Assumption of a single graphon per class in MomentMixup.

  - Lack of adaptation to dynamic or attributed graphs.

These are valid and well-stated. I would also add:

  - Lack of interpretability in the learned INR.

  - Training instability when motif estimates are noisy (especially for small motifs or sparse graphs).

**Quality:**

4

**Strengths And Weaknesses:**

**Strengths**:
1. *Novelty*: The idea of matching motif densities via an INR is new and elegant. It sidesteps latent position estimation—a common bottleneck in graphon estimation.

2. *Scalability*: The method is inherently scalable. It reduces both memory and computational overhead by discarding raw graphs after extracting moments and works in polynomial time.

3. *Theory-backed*: The paper offers a clear theorem bounding the cut distance between the estimated and true graphon under realistic assumptions (with number of motifs, sample size, etc.).

4. *Empirical Performance*: Strong results on synthetic graphon estimation (13 benchmark graphons) and real-world graph classification via MomentMixup. It outperforms or matches SOTA in most cases.

5. *MomentMixup is principled*: The paper identifies a limitation in G-Mixup (moment vectors of interpolated graphons ≠ interpolated moment vectors) and proposes a better method grounded in structure.

6. *Clear Writing and Presentation*: The exposition is mostly accessible and well-structured for a technical audience. Figure 2 and the breakdown of training steps help clarify the process.


**Weaknesses**:
1. *Choice of Moments is Fixed*: As admitted by the authors, the set of motifs is predefined (up to 5-node motifs via ORCA). There’s no adaptive selection or theoretical guidance on which motifs best characterize a given graphon. Poor motif choice can hurt performance.

2. *Single-Graphon Assumption for Each Class*: MomentMixup assumes each class corresponds to a single graphon. This is limiting for datasets with high intra-class variation (e.g., Reddit-Binary), where performance lags.

3. *No Comparison with Recent Diffusion-Based or Equivariant Methods*: SIGL is a strong baseline, but there are recent graph representation models (e.g., equivariant GNNs, diffusion-based models) that could offer complementary perspectives.

4. *No Ablation on INR Architecture*: The INR used is a shallow 1-layer MLP. It's unclear whether deeper architectures improve estimation quality or overfit due to noise in empirical moments. An ablation would be informative.

5. *MomentMixup’s Label Assignment*: The paper does not detail how the class labels for generated graphs via MomentMixup are assigned. Are they soft labels or nearest neighbor class labels? This has implications for supervised training.

---

> ### Author Rebuttal · Authors · 2025-07-31
>
> Thank you for your insightful comments and questions. We address those in the following subsections.
>
> **1. Choice of moments (Ablation Study):**
> Our method's robustness is demonstrated by its strong performance using a relatively small, fixed set of motifs. To investigate this further, we conducted an ablation study, with results for graphons 2 and 4 from Table 2 presented in the table below. These results highlight a key finding: while performance generally improves as more motifs are added, this trend is not strictly monotonic. For instance, we observed a minor dip in performance after incorporating the eighth motif before the trend resumed. A similar non-monotonic trend was also observed for Graphon 4.
> This behavior suggests that not all motifs contribute equally to the estimation; some are more informative, while others might introduce statistical noise, particularly higher-order motifs that require more samples for accurate approximation. An adaptive motif selection strategy could potentially optimize performance by identifying the most informative subgraphs for a given graphon, and we agree this is a valuable idea to explore.
> However, it is crucial to emphasize that these are minor fluctuations within a strong overall trend, and that adding motifs further than 6 does not provide a large advantage. The robustness of our current approach is confirmed by the fact that using a comprehensive set of motifs up to size $k$ consistently yields a superior estimation compared to using significantly fewer motifs. As shown in our tests on Graphon 2 and 4, even when performance dips slightly with the addition of a specific motif, the overall result remains highly effective. While statistical theory suggests all moments are needed to uniquely identify a distribution, our experiments affirm that a practical and powerful estimation can be achieved using a fixed set containing all motifs up to a small node count $k$ (note that 9 motifs are achieved with just $k=4$).
>
>
>
> | Motif | Avg GW (idx 2) | Std GW (idx 2) | Avg GW (idx 4) | Std GW (idx 4) |
> | :---: | :---: | :---: | :---: | :---: |
> | 1 | 0.089 | 0.012 | 0.151 | 0.009 |
> | 2 | 0.100 | 0.015 | 0.060 | 0.017 |
> | 3 | 0.036 | 0.008 | 0.037 | 0.012 |
> | 4 | 0.031 | 0.008 | 0.018 | 0.005 |
> | 5 | 0.029 | 0.008 | 0.017 | 0.005 |
> | 6 | 0.021 | 0.003 | 0.014 | 0.002 |
> | 7 | 0.024 | 0.003 | 0.018 | 0.004 |
> | 8 | 0.027 | 0.006 | 0.018 | 0.006 |
> | 9 | 0.026 | 0.002 | 0.013 | 0.003 |
> | 10 | 0.022 | 0.003 | 0.011 | 0.001 |
> | 11 | 0.023 | 0.005 | 0.012 | 0.002 |
> | 12 | 0.020 | 0.003 | 0.010 | 0.002 |
> | 13 | 0.020 | 0.002 | 0.010 | 0.001 |
> | 14 | 0.018 | 0.001 | 0.010 | 0.002 |
> | 15 | 0.020 | 0.002 | 0.010 | 0.001 |
>
>
> **2. Label assignment:**
> The labels are assigned using soft label computation, as detailed in Line 8 of the MomentMixup pseudocode (Algorithm 1) in the appendix. Intuitively, soft labeling provides a more accurate target by reflecting the proportional mix of the original samples. This process helps the model establish a clearer decision boundary between classes, improving its ability to generalize to new, unseen data.
>
> **3. INR Generalization:**
> To further test for generalization beyond the GW loss and centrality measures, we conducted an additional experiment. Following the experimental setup described in the paper, we trained a model on 9 motifs. We then evaluated its extrapolation performance on a different set of motifs, the 5-node subgraphs shown in the ORCA paper [1]. The results in the table below show that the moment estimations are highly accurate, indicating that the model successfully learned the true data distribution.
>
> | motif index | relative error(%) |
> | -- | -- |
> |            10 |                2.22421   |
> |            11 |                1.53611   |
> |            12 |                1.78493   |
> |            13 |                0.0728369 |
> |            14 |                0.204544  |
> |            15 |                1.03096   |
> |            16 |                1.89877   |
> |            17 |                0.461895  |
> |            18 |                0.153658  |
> |            19 |                0.371466  |
> |            20 |                1.74415   |
> |            21 |                1.50211   |
> |            22 |                1.10191   |
> |            23 |                2.46796   |
> |            24 |                0.139989  |
> |            25 |                1.04261   |
> |            26 |                3.08679   |
> |            27 |                1.75301   |
> |            28 |                2.06194   |
> |            29 |                4.01043   |
> |            30 |               11.0857    |
>
> **4. Heterogeneous graph classes:**
> We thank the reviewer for this insightful comment. We plan to address this important concern by extending our framework to handle heterogeneous classes, and our preliminary results are  promising.
>
> The key idea is to first partition the graphs within a class into structurally homogeneous clusters using their **moment vectors as features**. Graphs from the same generative mode will have similar moments, making them easily separable in this space. We then apply MomentNet to each cluster to learn a representative graphon for each distinct structural mode.
>
> We have conducted preliminary experiments confirming the effectiveness of this approach, and we will include a detailed analysis in the camera-ready version.
>
> This clustering-based strategy is a more explicit version of the principle already implicit in MomentMixup. Currently, by averaging moments different classes (Algorithm 1, Line 2), our method already mitigates some of this structural variance. This demonstrates the versatility of our moment-matching paradigm for modeling complex, multi-modal graph distributions.
>
> **5. Motif sampling efficiency:**
> To address this question, we highlight an insightful result from our scalability experiments in Figure 2. This figure demonstrates that estimating the true distribution does not require large graphs. Even when large graphs are available, they can be subsampled, as our method's performance remains robust in this small-graph regime. The runtime plots further affirm that using smaller graphs can yield a high-quality estimation efficiently. While subsampling can be applied to any method, Figure 2 shows that our approach has a distinct advantage in these settings, making it particularly practical.
>
> **6. Comparison with diffusion-based model:**
> The primary goal of our paper is to show that graphon estimation can be done without using a combinatorial loss function (Gromov-Wasserstein (GW) distance). To the best of our knowledge, there is no diffusion-based model that enables graphon estimation. It might be worth mentioning that our proposed method can be seen as a diffusion model with only a backward path, where we match the output distribution with our target based on moment matching.
>
> **7. INR ablation study:**
> We thank the reviewer for raising this important point and apologize for the lack of clarity in our experimental description. We did not, in fact, use a single architecture for all experiments. Our approach involved selecting an architecture appropriate for the complexity of the target graphon distribution. For simpler graphons, we used a shallow 1-layer MLP, as its limited capacity makes it robust against overfitting to potentially noisy empirical moments. For more complex graphons with high-frequency details, we employed a SIREN architecture, which is standard for INR tasks and is better suited to modeling intricate structures.
>
> This deliberate choice reflects the trade-off between model expressivity and regularization that the reviewer correctly identified. We recognize this was not made clear in the manuscript. To address this, we will update the experimental section in the camera-ready version to clearly state this as an hyperparameter choice.
>
> ---
>
>
> We hope these responses addresses your main concerns. Thanks once again for your constructive feedback and questions, which will strengthen the quality of the revised manuscript.
>
>
>
>
> ## References:
> [1] T. Hocevar, et al. (Bioinformatics 2014) A combinatorial approach to graphlet counting

---

> > ### Comment · Reviewer_ogzT · 2025-08-06
> >
> > Thank you to the authors for the detailed responses. I appreciate the additional clarifications and ablation studies provided. My concerns have been largely addressed, and I will consider raising my score.

---

> > > ### Author Response · Authors · 2025-08-06
> > > **Thank you for your thorough review**
> > >
> > > Thanks so much for the positive feedback! We're very happy that our response and the new experiments were helpful. We are grateful for your reconsideration.

---

### Note · Authors · 2025-08-15

**Author Final Remarks (summary of concerns & our responses)**

Thank you to the AC and the reviewers for the constructive discussion. Below is a concise recap of the main concerns and how we addressed them; all new analyses and experiments will be included in the camera-ready.

- **Motif selection & ablation.** We added an ablation across increasing motif sets (Graphons 2 & 4), showing gains saturated around all motifs up to 4 nodes (k=4) with mild non-monotonic dips, confirming robustness. We also tested extrapolation to unseen 5-node motifs and observed low relative errors, indicating the learned graphon generalized beyond the training dictionary. We will include guidance and discuss adaptive selection as future work.

- **Theorem 1 verification.** We computed the bounds under our experimental settings and found them non-vacuous in realistic regimes; our analysis explicitly connected the quantity of input data, the number of graphs and the number of nodes, to the final estimation error, empirically validated the accurate-motif-learning assumption, and confirmed that the theorem’s conditions hold in practice. We clarified Assumption 1 and showed the optimization-error proxy is small; we will add an appendix summarizing these findings.

- **MC gradient stability.** We measured gradient variance versus the number of Monte Carlo samples and observed consistent variance decay, supporting stable training.

- **Scalability & complexity.** We compared the worst-case ORCA bounds with our empirical measurements, which show near-cubic scaling (with a small constant) for 4-node counting on dense graphs; we also provided timing breakdowns and showed that MomentNet matched or exceeded SIGL runtime on large/dense settings while maintaining accuracy. We justified subsampling under the graphon model (multiple induced subgraphs are i.i.d. graphon samples), which preserved performance while reducing cost and serves as a practical strategy to manage runtime.

- **Dense graphs & large-scale settings.** Our benchmarks already included dense graphons; we conducted explicit =2k-node experiments during the rebuttal (details in the discussion) and will add them to the camera-ready, including a breakdown of counting vs. INR training and additional dense-regime results.

We believe these additions address the reviewers’ main concerns and strengthen both the empirical and theoretical foundations of MomentNet. Thanks again for the insightful feedback.

---

### Decision · Program_Chairs · 2025-09-17

**Decision:**

Accept (poster)

**Comment:**

This paper proposes MomentNet, a novel, scalable graphon estimation method using moment matching and implicit neural representations (INRs) of graphons. Given a set of graphs, all assumed to have been sampled from the same unknown ground-truth graphon, an INR representing a graphon is optimized to match the motif counts of the target graphs. The paper proposes a theorem for supporting the proposed method.

All reviewers evaluated this paper positively, stating that the paper is clear, the method is novel, has good numerical properties, and is backed by theory.

**Mandatory revision for camera-ready:** One significant shortcoming in the current presentation is that theorems are stated in the main paper without explicitly writing their assumptions, which creates a misleading impression of their significance. Once seeing the assumptions in the appendix, readers will realize that the theoretical results are weaker than they were initially led to believe. For example, **please move Assumption 1 to the main paper as it is a highly nontrivial condition of Theorem 1**, and hence not a small detail that can be ignored in the main paper. Moreover, the statement in Assumption 1 is in fact not formulated as an assumption. An assumption is a condition that you suppose on the parameters of the problem/theoretical setting. What you “assume” here is in fact a conjectured property. So change the title to a hypothesis or a conjecture, and move it to the main paper. Now, to make Theorem 1 rigorous, you need to formulate assumption 1 as a rigorous mathematical assumption on the parameters of the problem, deleting the sentence “Given a sufficiently expressive neural network architecture” which is an interpretation and not a mathematical statement. The assumption should simply be about the approximation error of the homomorphism densities. Hence, Theorem 1 should be clearly described as an approximation theorem, not an optimization theorem.

Moreover, please introduce all notations throughout the paper. For example, $\delta_M$ and $\eta$ are not defined in Theorem 1.